# scTWAS: a powerful statistical framework for single-cell transcriptome-wide association studies

**Zhaotong Lin** [1] **& Chang Su** [2,3] ✉

Transcriptome-wide association studies (TWAS) have successfully identified genes associated with complex traits and diseases, but most have been performed using bulk gene expression data, which aggregate signals across heterogeneous cell types. Population-scale single-cell RNA sequencing data now make it possible to perform TWAS at the cell-type resolution, but present unique challenges due to strong noises, technical variations, and high sparsity. Here, we propose scTWAS, a statistical method to conduct cell-type-specific TWAS using single-cell data. Leveraging a latent-variable model and moment-based estimation to address the challenges of single-cell data, scTWAS consistently improves the prediction of genetically regulated gene expression across cell types in both blood and brain tissues. Compared to existing methods, scTWAS identifies substantially more gene-trait associations across 29 hematological traits and three immune-related diseases in immune cell types. An application to Alzheimer's disease also reveals cell-subtype-specific associations, including *MS4A6A* in the disease-associated microglial subtype and *PPP1R37* in the inflammatory microglial subtype.

Genome-wide association studies (GWAS) have successfully identified numerous genetic variants associated with complex traits and diseases. However, the underlying biological mechanisms driving these associations remain unclear, as the majority of risk variants lie in non-coding regions and may exert their effects through gene regulation[1–3]. Transcriptome-wide association studies (TWAS) address this challenge by integrating GWAS data with gene expression to identify genes whose genetically regulated expression (GReX) is associated with the trait of interest[4–6]. It consists of two stages, which first trains a prediction model on the genetic regulation of gene expression with a reference panel of paired genotype and transcriptome data. In Stage 2, these prediction models are applied to GWAS data (either individual-level genotype and phenotype data or GWAS summary statistics) to test for associations between predicted gene expression and the trait of interest. Applications of TWAS have yielded numerous findings on the biological functions underlying disease pathogenesis and have further prioritized concrete targets for intervention and treatment[7–13].

Most TWAS to date use reference transcriptome data collected from disease-related tissues, such as peripheral blood mononuclear cells (PBMC) for immune diseases[13,14] and brain tissues for Alzheimer's disease (AD)[7,8], and prioritize genes whose tissue-level GReX is associated with complex diseases. However, tissues consist of biologically heterogeneous cell types, among which the genetic regulation of gene expression differ. For example, expression quantitative trait loci (eQTL) studies have demonstrated distinct genetic regulation across cell types in PBMC[15–17] and in brain[18]. Furthermore, specific cell types may contribute more significantly than others in disease pathogenesis. For example, heritability enrichment analysis shows that microglia carries a sevenfold higher genetic risk for AD compared to other brain cell types[19]. Beyond cell types, recent studies also highlight the importance of investigating cell subtypes to better understand disease mechanisms, where cell subtypes represent more refined groups within a cell type defined by distinct gene expression signatures. For instance, a microglia subtype, known as disease-associated microglia,

[1]Department of Statistics, Florida State University, Tallahassee, FL, USA. [2]Department of Biostatistics and Bioinformatics, Emory University, Atlanta, GA, USA. [3]Department of Human Genetics, Emory University, Atlanta, GA, USA. ✉e-mail: chang.su@emory.edu

contributes uniquely to restricting neurodegeneration and has a higher expression of AD risk genes compared to other homeostatic microglia[20,21] with potentially subtype-specific genetic regulation[18]. In contrast, most existing TWAS use reference gene expression data collected on bulk tissues, where mRNA molecules from heterogeneous cell types have been aggregated during data collection. As a result, they may fail to capture genetic regulation and disease associations specific to cell types and subtypes that play critical roles in diseases.

Conducting TWAS at the level of cell types is essential for elucidating the molecular and cellular mechanisms of diseases. While this could be performed with bulk data collected on sorted cell populations, the heterogeneity across cell subtypes may be obscured, as cell sorting primarily relies on well-established markers for major cell types, and the sample size may be insufficient for well-powered statistical analysis[16,22]. The recently available population-scale single-cell RNA-sequencing data (scRNA-seq) provide an unprecedented opportunity to address these limitations. These data profile hundreds of individuals, each with thousands of cells per individual in disease-related tissues, identifying a dozen of major cell types and tens of refined cell subtypes [e.g., refs. 17,18,23,24]. This enables the modeling of genetic regulation and the identification of disease-associated genes across diverse cellular contexts.

However, the unique characteristics of scRNA-seq data, including strong noises, technical variations, and high sparsity, challenge existing statistical methodologies proposed for TWAS. Count data from scRNA-seq measure underlying gene expression levels in single cells with strong noises and technical variations from the sequencing experiments[25]. If left unaddressed, these measurement noises would attenuate the estimates of genetic effects, while technical variations across cells introduce artificial differences in mean expression and heteroskedastic variance, even among cells with the same underlying gene expression levels. The counts are also highly sparse due to the limited starting materials in single cells and the low capture efficiency of the technology[26]. In contrast, most existing TWAS methods were developed for bulk RNA-seq data and are therefore better suited to normalized and continuous expression traits with relatively homoskedastic variance [e.g., refs. 4,5]. This has motivated recent cell-type-specific TWAS analysis to apply heuristic normalization to single-cell data before applying existing TWAS methods[18,27–30]. However, these normalization strategies may lack theoretical justification and fail to account for the unique characteristics of single-cell data[25,26,31], resulting in reduced accuracy in learning the genetic regulation of gene expression and diminished power to detect associations between GReX and diseases.

To address these challenges, we propose scTWAS, a statistical method for conducting cell-type-specific TWAS with scRNA-seq data. scTWAS introduces a latent-variable model that simultaneously models the effects of genetic regulation on the underlying gene expression and the measurement noises and technical variations introduced by sequencing experiments. By disentangling biological gene expression from technical noises, this model enables accurate quantification and estimation of GReX. To improve statistical efficiency under this model, we further introduce a variance-weighted moment-based regression that explicitly accounts for the heteroskedastic noise in estimation. These designs lead to improved accuracy of GReX prediction and higher statistical power of identifying TWAS genes in cellular contexts of interests with scRNA-seq data.

We applied scTWAS on multiple population-scale scRNA-seq datasets to construct GReX prediction models and test the association between predicted GReX and traits in relevant cellular contexts. Our results demonstrate that scTWAS consistently outperformed existing approaches in GReX prediction across cell types and independent datasets. The improvement is particularly pronounced in settings with fewer cells, where GReX is more difficult to predict. Using GReX models for immune cell types, scTWAS systematically identified more associations with hematological traits and uncovered cell-type-specific regulatory mechanisms underlying genes linked to immune-related diseases, providing new findings that have not been previously reported by GWAS or TWAS with bulk gene expression data. Using GReX models for brain cell types, scTWAS revealed the cellular contexts in which both established and previously unreported genes may contribute to AD pathogenesis. Notably, it uncovered regulatory mechanisms of *MS4A6A* specific to a subtype of disease-associated microglia that are not shared with surveilling or reactive microglia, underscoring the importance of refined cellular resolution and single-cell-based TWAS for understanding the functional impact of disease-associated variants. The GReX prediction models for immune and brain cell types are made publicly available ("Data availability").

## Results
### Method overview

scTWAS consists of two stages: Stage 1 constructs cell-type-specific GReX prediction models for each gene using the genotype data and single-cell gene counts from a specific cell type; and Stage 2 tests for the association of predicted GReX and a trait of interest using the trained Stage 1 models and GWAS summary statistics or individual-level data.

In Stage 1, we model pseudo-bulk counts, defined as the total gene counts across all single cells per cell type and individual, in order to mitigate the high sparsity in single cells and to represent individual-level gene expression. For a given gene-cell type pair, let $z_i$ denote the underlying expression level, defined to be the relative abundance of this gene (i.e., the number of mRNA molecules from this gene relative to the total number of molecules) from this cell type in individual $i$. Let $\mathbf{g}_i$ be a vector of *cis*-genotype (*cis*-single nucleotide polymorphisms, SNPs) in individual $i$ and $\mathbf{c}_i$ be a vector of covariates (e.g., intercept, sex, age, genotype principal components). We assume that

$$z_i \sim F(\mu_i, \sigma_i^2), \ \mu_i = \mathbf{g}_i'\boldsymbol{\beta} + \mathbf{c}_i'\boldsymbol{\gamma}, \qquad (1)$$

where $F(\mu_i, \sigma_i^2)$ denotes a nonnegative distribution with mean $\mu_i$ and variance $\sigma_i^2$, and $\boldsymbol{\beta}$ and $\boldsymbol{\gamma}$ denote the coefficients for genotypes and covariates, respectively. This assumes that the underlying gene expression is a random variable whose expectation depends on genotypes and covariates, and whose variance reflects the biological variations across individuals. It makes no parametric assumption on the distribution of $z_i$, accommodating common assumptions such as Gamma distribution[26,32]. We call this the expression model, which models the genetic regulation of gene expression in the cell type. In practice, $z_i$ is a latent-variable to be measured by sequencing experiments. To model the measurement process, we use $x_i$ to denote the observed pseudo-bulk counts from individual $i$ and $s_i$ to denote the sequencing depth for individual $i$, defined as the total counts across all genes from individual $i$ in the cell type of interest. We propose the following measurement model:

$$x_i|z_i \sim \text{Poisson}(s_i z_i). \qquad (2)$$

This assumes that the observed gene counts are measured with Poisson sampling noises[25,26], whose magnitude further depends on sequencing depths, a technical factor known to substantially vary across samples[26]. Together, (1) and (2) give the expression-measurement model[25,31], which accurately quantifies the genetic regulation of underlying gene expression with $\mathbf{g}_i'\boldsymbol{\beta}$ while accounting for noises and technical variations in measured gene counts. To build GReX prediction models (i.e., $\mathbf{g}_i'\boldsymbol{\beta}$), we propose a penalized moment-based regression to estimate $\boldsymbol{\beta}$, where an iteratively re-weighted least squares (IRLS) method is employed to downweight noisy gene counts with large variance and improve statistical efficiency. Details are presented in Section "scTWAS method".

In Stage 2, we test associations between predicted GReX and the trait of interest in the cell type of interest. When individual-level GWAS data (both genotype and phenotype) are available, cell-type-level gene expression can be directly predicted using SNP coefficients estimated from Stage 1 (i.e., $\hat{\boldsymbol{\beta}}$), and its association with the phenotype can be tested using regression methods[4]. In most cases where individual-level data are not available, we can perform the test using GWAS summary statistics and a linkage disequilibrium (LD) reference panel[5] (Section "scTWAS method").

In summary, scTWAS integrates single-cell data with genetic information to identify cell-type-specific gene-trait associations. By leveraging an expression-measurement model and a moment-based estimation framework, scTWAS effectively accounts for noises and variations from the measurement process and models the genetic regulation of underlying expression levels, eliminating the need for heuristic normalization. It provides more accurate GReX predictions, which then improves the statistical power for association testing with complex traits. This method enhances our ability to uncover biologically meaningful insights at the cell-type-level, advancing our understanding of molecular and cellular mechanisms of complex diseases.

## scTWAS achieves higher prediction accuracy and power on simulated data

To evaluate the Stage 1 prediction performance and Stage 2 power of scTWAS, we simulated realistic single-cell gene expression data based on the characteristics of OneK1K study[17], which is one of the first population-scale scRNA-seq datasets, comprising 1.27 million cells from 14 immune cell types across 982 individuals. We evaluated the performance of scTWAS against two representative approaches used in recent cell-type-specific TWAS analyses with scRNA-seq data: NA-TWAS[29] and AN-TWAS[28]. Both approaches used an elastic net model for Stage 1 prediction, but applied different normalization steps to single-cell data. Briefly, NA-TWAS first normalized the single-cell count data before aggregating it based on the suggestion of a recent benchmarking paper[33], while AN-TWAS aggregated the single-cell count data first and applied normalization on the pseudo-bulk count (Supplementary Fig. 1, see Section "Other TWAS methods under comparison"). We simulated GReX with parameters estimated on real data, and the observed single-cell counts based on an expression-measurement model[25,26,31] (see Section "Simulation settings"), which captured the sparse and heteroskedastic nature of scRNA-seq data, allowing for a realistic evaluation of method performance. We selected six cell types (CD4$_{NC}$, CD8$_{ET}$, B$_{IN}$, Mono$_C$, NK$_R$, and Plasma), each representing different levels of cell type abundance in real data, with a mean of 472, 209, 84, 40, 10, and 5 cells per individual, respectively. See Section "Genotype and scRNA-seq data" for the definition of cell types.

We first performed one-sample individual-level simulations where both single-cell gene expression and trait phenotypes were generated at the individual-level using the OneK1K cohort (Section "One-sample individual-level TWAS"). Our results demonstrate that scTWAS consistently outperformed NA-TWAS and AN-TWAS in both prediction accuracy and power across cell types with varying abundance. We evaluate Stage 1 GReX prediction models by the prediction $p$-value from regressing observed expression on the estimated GReX in a fivefold cross-validation (see Section "Prediction model training and evaluation"). Figure 1a shows that scTWAS yielded a greater number of significant Stage 1 GReX prediction models compared to AN-TWAS and NA-TWAS (nominal $p$-value < 0.05). In Fig. 1b, scTWAS also exhibited higher power to detect gene-trait associations than the other two approaches, with the extent of the improvement varying by cell type abundance. Notably, the gain in power was more pronounced in less abundant cell types such as B$_{IN}$ and Mono$_C$ (0.23 and 0.30 higher than AN-TWAS, respectively). This highlights scTWAS's ability to effectively leverage sparse data and enhance power in challenging scenarios, which are common among immune cell types in the OneK1K dataset,

where 9 out of 13 have abundances less than or equal to B$_{IN}$ (Supplementary Data 1). Even in highly sparse cell types such as NK$_R$ and Plasma, where all methods exhibited low power, scTWAS still maintained an advantage. These results suggest that scTWAS is more powerful in uncovering gene-disease association across cellular contexts of varying abundance, with particularly strong gains in disease-relevant but less abundant cell types and subtypes. This includes microglia (subtypes) for AD as an example, which will be further demonstrated in Section "Cell-subtype-level scTWAS analysis identifies AD-gene associations in microglia subtypes".

To mimic typical applications, we also performed two-sample simulations in which Stage 2 uses GWAS summary statistics independent of the gene expression cohort (Section "Two-sample TWAS with GWAS summary data"). As shown in Fig. 1c, we considered two expression reference panel sample sizes, $N_{eQTL} \in \{500, 982\}$, where $N_{eQTL}$ denotes the number of individuals in the expression reference panel used to train the stage 1 GReX model, and three different GWAS sample sizes, $N_{GWAS} \in \{3 \times 10^4, 5 \times 10^4, 10^5\}$. As expected, power increased with larger expression reference panel sample size and with larger GWAS sample size. Across all settings, scTWAS remained the most powerful method. For the rare NK$_R$ cell type, however, all methods exhibited low power when the gene expression sample size is 500, reflecting the limited predictive performance of the Stage 1 models (Supplementary Fig. 2).

We evaluated the methods' type-I errors in two scenarios: one where gene expression was independent of the *cis*-SNPs (i.e., a null Stage 1), and another where the phenotype was independent of the gene expression (i.e., a null Stage 2). All methods, including scTWAS, successfully controlled the type-I error rates across all cell types (Supplementary Table 1, Supplementary Fig. 3). Under a null Stage 1, all methods yielded out-of-sample $R^2$ close to 0 as expected (Supplementary Fig. 4).

We also performed additional simulations to evaluate the three methods in diverse scenarios. First, we varied per-gene trait heritability (Supplementary Fig. 5). As expected, power increased as a gene explained a larger proportion of trait heritability. Second, we considered a sparser genetic architecture with three causal *cis*-SNPs (Supplementary Fig. 6), which reduced Stage 1 predictability and, in turn, Stage 2 power. Third, we introduced LD reference mismatch at Stage 2 by using an out-of-sample LD matrix (Supplementary Section 1.3.1), where results were similar to the matched-LD setting (Supplementary Fig. 7). Across these settings, scTWAS consistently achieved the highest power. Finally, when we introduced horizontal pleiotropy (Supplementary Section 1.3.2), all three methods exhibited inflation (Supplementary Figs. 8 and 9), consistent with the well-known sensitivity of standard TWAS to pleiotropic effects. We note this limitation and discuss potential extensions in the "Discussion".

## scTWAS enhances GReX prediction across cell types in blood and brain tissues

To benchmark GReX prediction models on real data, we applied scTWAS, NA-TWAS, and AN-TWAS to build cell-type-specific models using two population-scale single-cell datasets on PBMC[17] and brain tissues[18], and evaluated their performance based on both within-study and cross-study prediction accuracy ("Methods"). In the first application, we used OneK1K genotype and scRNA-seq data to build prediction models for 13 immune cell types. The average number of cells per individual varied widely across cell types, from 472 in CD4$_{NC}$ to 5 in Plasma, with a median of 49 (Supplementary Data 1). We built prediction models for genes whose total counts in the cell type are greater than 1000.

We assessed the accuracy of scTWAS, NA-TWAS, and AN-TWAS GReX prediction models using the prediction $p$-value in a fivefold cross-validation (see Section "Prediction model training and evaluation"). As shown in Fig. 2a, scTWAS identified more significantly

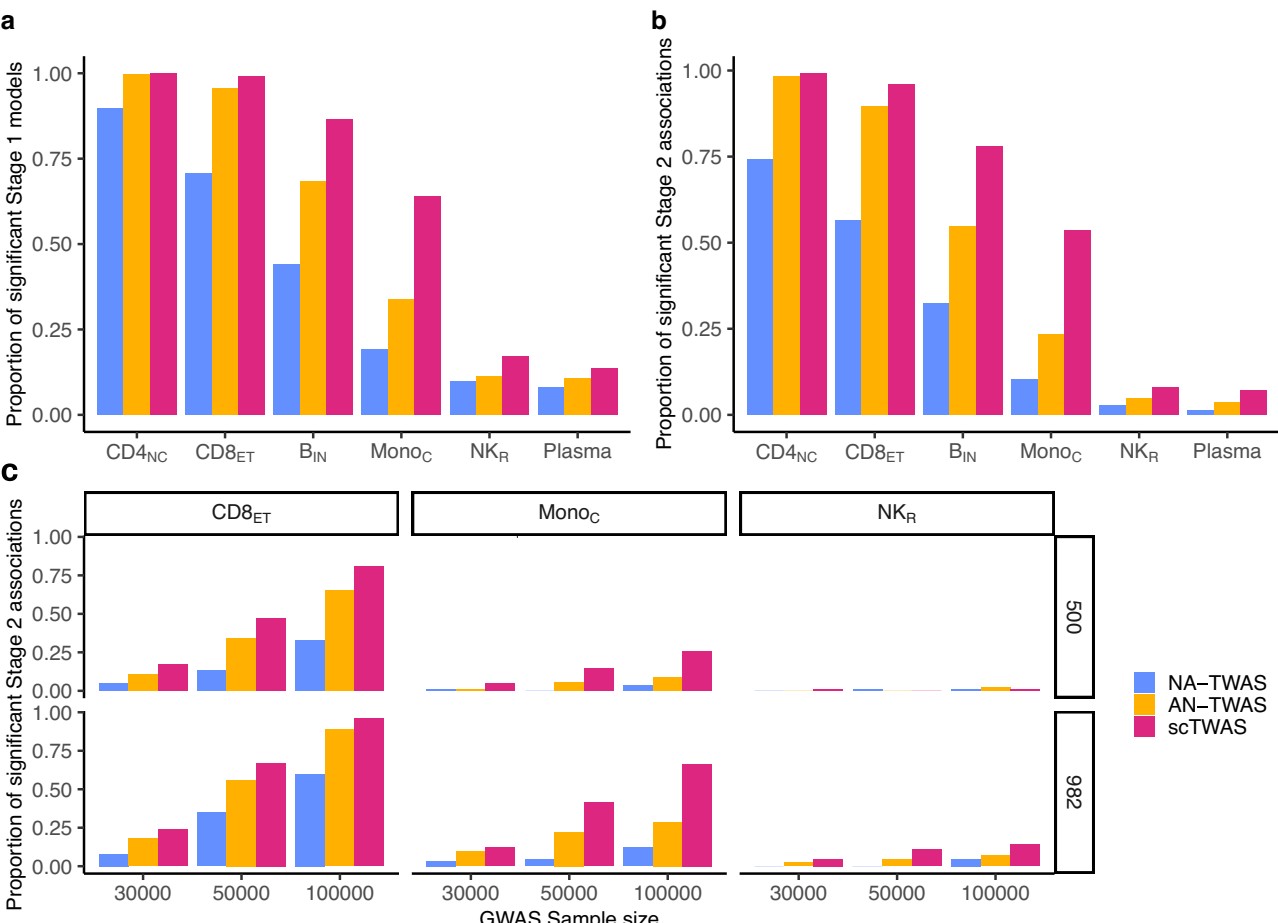

**Fig. 1 | Simulation results.** Performance evaluation of scTWAS, AN-TWAS, and NA-TWAS in one-sample individual-level simulations (**a**, **b**) and two-sample simulations with GWAS summary statistics (**c**) based on OneK1K. **a** Proportion of predictive GReX models (nominal *p*-value < 0.05). The nominal *p*-values are from a two-sided Wald test of the slope coefficient in a linear regression of observed gene expression on predicted gene expression, evaluated on validation folds within a cross-validation framework. **b** Empirical power to detect gene-trait associations across cell types of varying abundances when the gene's *cis*-SNPs explained 0.05 of the trait variance. **c** Empirical power across $N_{eQTL} \in \{500, 982\}$ and

$N_{GWAS} \in \{3 \times 10^4, 5 \times 10^4, 10^5\}$ when the gene's *cis*-SNPs explained 0.1/180 of the trait variance. In (**b**, **c**), two-sided *p*-values are from the Stage 2 association test and are adjusted for multiple testing within each cell type. The numbers of tests are: $CD4_{NC}$ (2189), $CD8_{ET}$ (1234), $B_{IN}$ (580), $Mono_C$ (476), $NK_R$ (122), and Plasma (91), corresponding to the numbers of significant Stage 1 genes within each cell type based on the real data. $CD4_{NC}$, $CD8_{ET}$, $B_{IN}$, $Mono_C$, $NK_R$, and Plasma have a mean of 472, 209, 84, 40, 10, and 5 cells per individual, respectively. Colors represent different methods being compared.

predicted genes across cell types, followed by AN-TWAS and NA-TWAS (Fig. 2b, Bonferroni corrected *p*-values < 0.05). To further demonstrate the cross-study prediction performance of scTWAS, we applied the prediction models trained on OneK1K to predict cell-type-specific bulk gene expression data from the database of immune cell expression (DICE, eQTL, and epigenomics) project[16] for six immune cell types that match between two independent studies ("Methods"). Figure 2b shows that scTWAS generated a median of 16% and 32% more significantly predicted genes compared to AN-TWAS and NA-TWAS, respectively (nominal *p*-values < 0.05 for Pearson correlation, Supplementary Fig. 10).

In addition to immune cell types, we further benchmarked scTWAS on GReX prediction models for brain cell types. We leveraged data from the Religious Orders Study and Memory and Aging Project (ROSMAP) study, which collected multi-omics data on postmortem brain samples, including paired whole genome sequencing data[34] and single-nucleus RNA-seq (snRNA-seq) data on dorsolateral prefrontal cortex across 424 individuals[18]. The nuclei have been annotated at two nested levels: major brain cell types and their subtypes[18,21]. Accordingly, we built prediction models for 6 brain cell types and 75 subtypes ("Methods"). The abundances of major brain

cell types range from 1403 cells per individual in excitatory neurons to 135 cells per individual in oligodendrocyte precursor cells (OPC) with a median of 525 cells per individual. For subtypes, the median abundance is 29 cells per individual (SD = 58) (Supplementary Data 1).

Figure 3a compares the prediction *p*-values in a fivefold cross-validation for microglia, demonstrating that scTWAS outperformed the other two methods across genes in generating more significant GReX models. To further validate prediction performance across studies, we applied the trained models to an independently collected snRNA-seq dataset on frontal cortex with 2.3 million nuclei from 427 individuals from the ROSMAP study, where 396 individuals have paired genotype data available[35]. As shown in Fig. 3b, scTWAS again achieved more significant prediction *p*-values compared to the existing methods across the six major cell types. Furthermore, at the subtype level (Fig. 3c and Supplementary Fig. 18), scTWAS identified markedly more predictable genes than the other methods across the 75 subtypes, with the largest gains observed in microglia subtypes. This highlights the advantages of scTWAS in predicting GReX for cellular contexts with sparser data and fewer cells. Taken together, these results on OneK1K and ROSMAP studies (Figs. 2 and 3) demonstrate that scTWAS

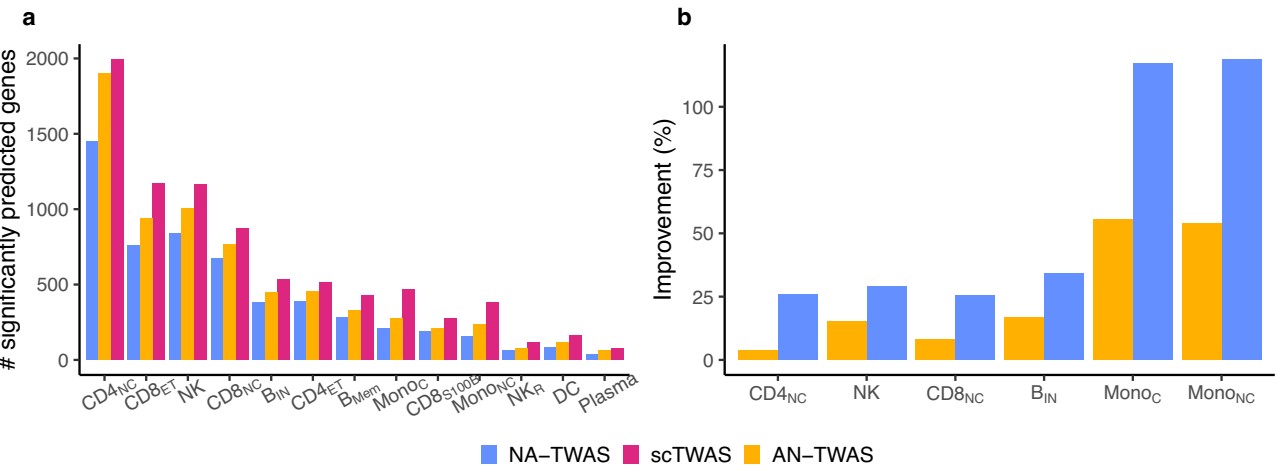

**Fig. 2 | Improvement of scTWAS over existing methods in Stage 1 GReX prediction models for immune cell types. a** Number of significantly predicted genes by three methods on the OneK1K dataset. **b** Percentage increase in the number of significantly predicted genes by scTWAS compared to NA-TWAS and AN-TWAS, evaluated on the independent DICE validation dataset. Colors represent different methods being compared.

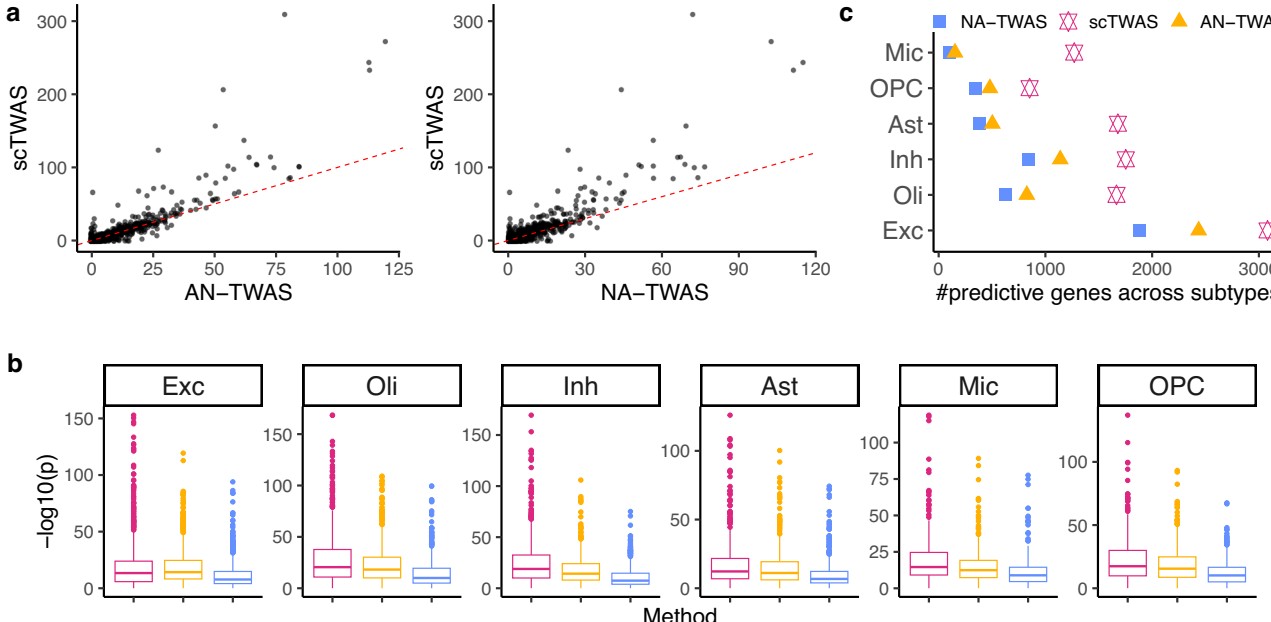

**Fig. 3 | Improvement of scTWAS over existing methods in GReX prediction models trained on the ROSMAP datasets. a** Comparison of $-\log_{10}$ $p$-values of GReX prediction between scTWAS and AN-TWAS (left), NA-TWAS (right) on microglia from 424 individuals, evaluated within the same study[18] using a cross-validation framework. **b** $-\log_{10}$ $p$-values of GReX prediction across studies, using models trained with ref. 18 and tested on an independently generated snRNA-seq dataset from frontal cortex with 396 individuals[35]. Colors represent different methods being compared, see legend in (**c**). The unadjusted $p$-values in (**a**, **b**) are from a two-sided Wald test of the slope coefficient in a linear regression of observed gene expression on predicted gene expression. Boxplots display the median (center), the first to the third quartiles (box), and whiskers extending to values within 1.5× the distance between the quartiles. Points indicate outliers beyond this range. The number of observations in each box corresponds to the number of genes evaluated: Exc (1159), Oli (683), Inh (609), Ast (569), Mic (301), and OPC (386). **c** Number of significantly predicted genes by Stage 1 prediction models for brain cell subtypes, aggregated across subtypes to their corresponding major cell types. Excitatory neuron (Exc), oligodendrocyte (Oli), inhibitory neuron (Inh), astrocyte (Ast), microglia (Mic), OPC have a mean of 1404, 748, 556, 495, 181, 135 cells per individual, respectively. Details of the number of cells and individuals are provided in Supplementary Data 1.

consistently generates better GReX prediction across tissue types and across datasets, which lays the foundation for increased power in testing gene-trait associations.

Additionally, to evaluate the gain by cell-type-specific TWAS compared to analyses that do not resolve cell types, we consider bulk-like TWAS analyses in which single cells are aggregated across all cell types to approximate bulk tissue samples, thereby enabling a

controlled comparison using the same single-cell data (Section "Prediction model training and evaluation"). On average, the bulk-like TWAS achieved higher prediction accuracy, which is expected given the larger sequencing depth and the reduced sparsity of aggregated expression data. However, cell-type-specific TWAS improved prediction accuracy for a subset of genes and uncovered regulatory patterns that were obscured in the bulk analysis

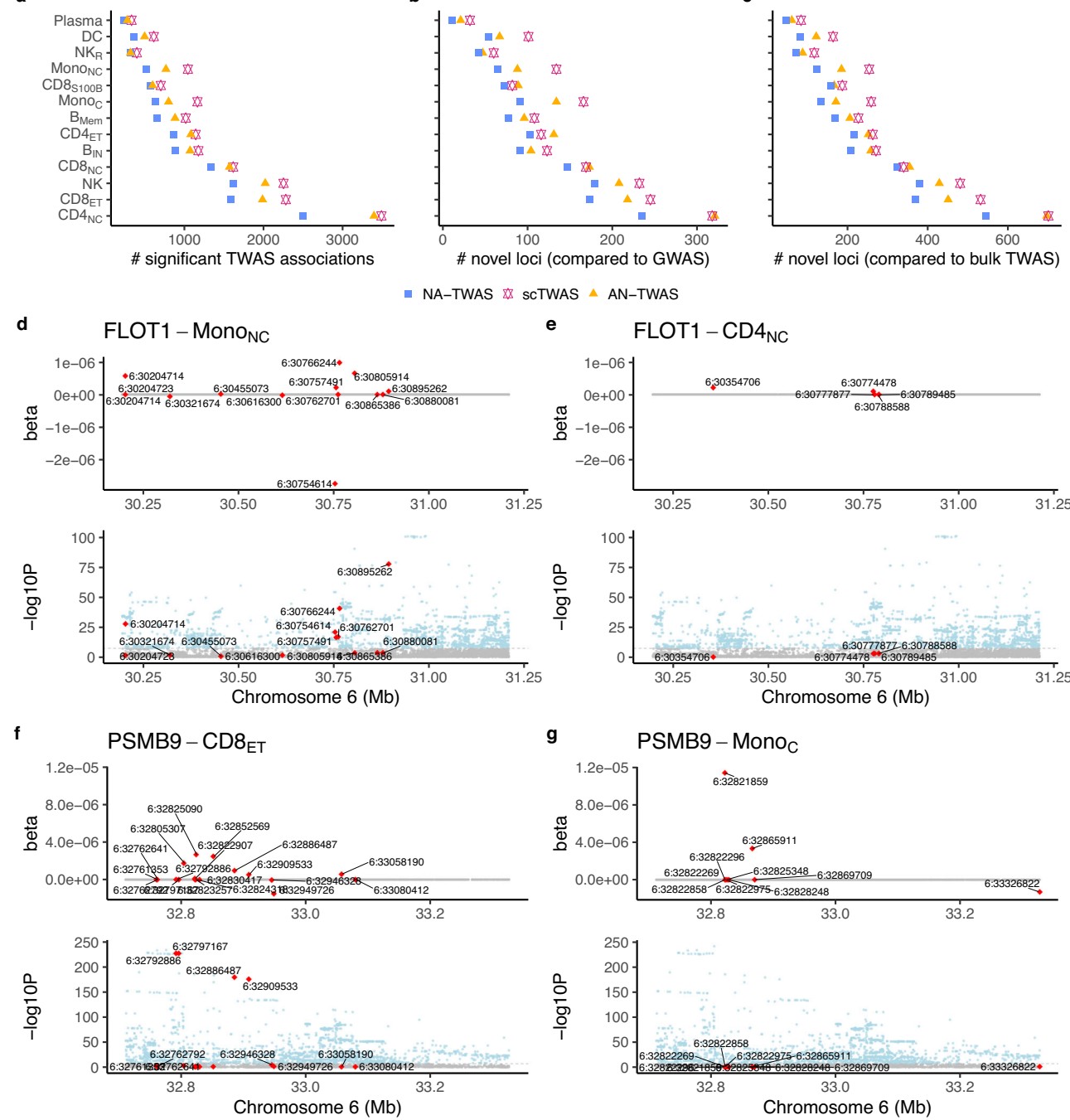

**Fig. 4 | scTWAS results of 29 hematological traits and three immune diseases using GReX prediction models trained with OneK1K scRNA-seq data. a** Total number of significant TWAS associations across 29 hematological traits by three methods. **b** Number of novel loci compared to GWAS. **c** Number of novel loci compared to whole blood TWAS[36]. Colors represent different methods being compared. **d**, **e** Top: Stage 1 model trained by scTWAS of *FLOT1* in Mono$_{NC}$ and

CD4$_{NC}$. Bottom: Rheumatoid arthritis GWAS association *p*-values around *FLOT1*. **f**, **g** Top: Stage 1 model trained by scTWAS of *PSMB9* in CD8$_{ET}$ (left) and Mono$_C$ (right). Bottom: Rheumatoid arthritis GWAS association *p*-values around *PSMB9*. In (**d**–**g**), SNPs whose coefficients in GReX prediction models are non-zero are colored in red; the two-sided *p*-values are extracted from the GWAS summary statistics without multiple testing adjustment.

(Supplementary Figs. 13 and 14a), with scTWAS yielding the largest number of unique findings and underscoring the value of cell-type-specific analyses.

## scTWAS discovers more associations with blood cell traits in immune cell types

Building on the improved Stage 1 performance of scTWAS on real data, we next benchmark its power to detect gene-trait associations in Stage 2 for hematological traits. Using the prediction models for immune cell

types from Section "scTWAS enhances GReX prediction across cell types in blood and brain tissues", we performed cell-type-level TWAS of 29 hematological traits with GWAS summary statistics ("Methods"). Transcriptome-wide significance was determined using the Bonferroni corrected *p*-value threshold to account for the total number of genes tested in each cell type. Figure 4a compares the total number of significant gene-trait associations identified by three methods across cell types (also see Supplementary Fig. 11 for a breakdown by individual GWAS). Consistent with the Stage 1 results, scTWAS identified the most

associations across cell types, followed by AN-TWAS and NA-TWAS, and more significant gene-trait associations were identified in more abundant cell types. We further classified the significant genes into loci ("Methods"). If a locus harbors a genome-wide significant GWAS variant, it is considered known; otherwise, it is considered novel. scTWAS increased the total number of loci by an average of 17% and 45% across cell types compared to AN-TWAS and NA-TWAS, respectively, including 18% and 56% more novel loci (Fig. 4b). The most substantial increases were observed in monocytes (both classical and non-classical), with the relative increment in the number of loci being up to 41% compared to AN-TWAS and 95% compared to NA-TWAS.

Additionally, we compared our cell-type-level TWAS results with bulk TWAS on the same traits[36], which used bulk RNA-seq data from whole blood on 922 European individuals to train the GReX models. As shown in Fig. 4c, cell-type-level TWAS uniquely identified novel loci that were missed by tissue-level TWAS, with scTWAS identifying the greatest number of novel loci overall. For example, *HHEX* is a member of the homeobox gene family, which plays an important role in embryogenesis and hematopoietic progenitor development[37]. While both tissue-level and cell-type-level TWAS identified its association with mean platelet volume, eosinophil count, eosinophil percentage, and lymphocyte percentage, the cell-type-level TWAS revealed that these associations were specific to $CD8_{ET}$ and NK cells. This cell-type specificity underscores the unique value of cell-type-level TWAS in pinpointing the relevant cellular contexts where gene expression influences blood cell traits, providing more detailed and biologically meaningful insights than tissue-level analyses. Furthermore, scTWAS uniquely identified an association between *HHEX* expression and monocyte percentage in $CD8_{ET}$ cells, which was not identified by NA-TWAS and AN-TWAS. These findings suggested a potential broader role for *HHEX* in hematopoiesis and immune cell regulation, consistent with its previously reported functions[38,39].

Another example is the *DGCR6* locus, where scTWAS uniquely identified its association with mean platelet volume and platelet count in $Mono_C$, an association that was not discovered with AN-TWAS, NA-TWAS, or bulk level TWAS. scTWAS revealed that the GReX of *DGCR6* was positively associated with platelet count and negatively associated with mean platelet volume, corroborating recent findings that increased *DGCR6* expression was implicated in reducing mean platelet volume[40]. Furthermore, *DGCR6*, located in the 22q11.2 region, is a candidate for involvement in DiGeorge syndrome (22q11.2 deletion syndrome) pathogenesis[41]. Individuals with 22q11.2 deletion syndrome have been shown to exhibit lower platelet counts and increased mean platelet volume[42]. Our cell-type-specific findings may offer new insights into the regulatory roles of *DGCR6*, particularly in how its expression in $Mono_C$ might influence platelet dynamics, potentially informing the mechanisms underlying 22q11.2 deletion syndrome. We also compared with the bulk-like TWAS constructed from the same OneK1K dataset (Section "Prediction model training and evaluation"), which confirmed the complementary role of cell-type-specific TWAS relative to bulk TWAS by identifying associations that may be missed in bulk analyses, with scTWAS yielding the largest number of new findings (Supplementary Fig. 15).

## scTWAS reveals mechanistic insights into immune-mediated diseases in immune cell types

Rheumatoid arthritis (RA), systemic lupus erythematosus (SLE), and asthma are complex immune-mediated diseases that collectively affect millions of individuals worldwide and pose a significant public health burden[43–45]. GWAS have identified hundreds of risk loci associated with these conditions[46–48], underscoring their highly polygenic architectures, but the functional interpretation of these loci remains limited due to the complexity of regulatory mechanisms and cellular heterogeneity in disease-relevant tissues. Immune cell types play a central role in the etiology of these diseases, as T cells, B cells, plasma cells,

and monocytes have all been implicated through functional and transcriptomic studies, highlighting the importance of cell-type-specific analysis in understanding disease mechanisms[23,24,49].

Here, we performed cell-type-level TWAS on these three immune-related diseases using cell-type-specific GReX models trained on OneK1K immune cell data from Section "scTWAS enhances GReX prediction across cell types in blood and brain tissues". In this analysis, scTWAS identified 64, 41, and 111 genes associated with RA, SLE, and asthma, respectively, in at least one of the 13 immune cell types. This corresponded to 11, 15, and 34 more genes than those identified by NA-TWAS, and 6, 6, and 16 more than those identified by AN-TWAS. Moreover, scTWAS uniquely identified 8 genes for RA, 11 for SLE, and 29 for asthma that were not identified by either NA-TWAS or AN-TWAS. We highlight a few examples here, while a complete list of significant genes is provided in Supplementary Tables 3–5. scTWAS uniquely identified an association between *IRF5* and RA and SLE in $Mono_C$ and NK cells, uncovering its cell-type-specific roles in disease mechanisms. *IRF5* is a well-documented transcription factor critical in the regulation of immune responses, particularly in the production of type I interferons, and has been proposed as a therapeutic target for autoimmune diseases[50]. Previous studies have shown that *IRF5* mediates joint inflammation by integrating Toll-like receptor signals and regulating proinflammatory cytokine and chemokine production in a mouse model[51,52], and its inhibition suppresses SLE onset and progression in mice[53,54]. Tissue-based TWAS also identified its association with RA and SLE[12,55]. Our results corroborate and extend prior evidence by identifying specific immune cell types, $Mono_C$ and NK cells, through which *IRF5* may contribute to SLE pathogenesis. These results support further investigation of *IRF5*, particularly its inhibition in $Mono_C$ and NK cells, as a promising direction for SLE drug discovery. scTWAS also uniquely identified *SULF2* as significantly associated with asthma in plasma cells. Known for its role in modifying heparan sulfate proteoglycans, *SULF2* can impact key biological processes such as cell signaling, inflammation, and tissue remodeling, including airway remodeling, a hallmark of chronic asthma[56]. Notably, while genetic variants in *SULF2* gene have been associated with several diseases, there is currently no reported association with asthma or other lung diseases. Our findings provide evidence linking *SULF2* to asthma pathogenesis, specifically through its potential role in plasma cells, and highlight it as a candidate for future functional studies and therapeutic exploration. Furthermore, a comparison with the bulk-like TWAS showed that scTWAS identified dozens of associations that were not observed in the bulk-like analysis (Section "Prediction model training and evaluation", Supplementary Fig. 15).

Next, we investigated the similarity and specificity of scTWAS findings across immune cell types. 21 out of 64 RA genes identified by scTWAS were specific to a single-cell type, while the majority were shared across two or more cell types (also see Supplementary Fig. 16 for a detailed UpSet plot). Several possibilities could lead to the cell-type specificity observed in scTWAS results. First, a gene may be expressed exclusively in one cell type. For example, *ZFP57* was highly expressed (total counts > 1000) only in $CD4_{NC}$ cells and was associated with RA in that cell type. Second, a gene may be expressed in multiple cell types, but a predictive GReX model could be obtained only in a specific cell type, potentially due to cell-type-specific genetic regulation. For example, *FLOT1* was expressed in all 13 immune cell types, but only the scTWAS GReX model in $Mono_{NC}$ was predictive, and significant associations with RA, SLE, and asthma were observed only in this cell type. Figure 4d shows the estimated genotypes' effects by scTWAS for *FLOT1* in $Mono_{NC}$ with a prediction p-value of $4.3 \times 10^{-12}$. In contrast, the prediction p-value is only 0.36 in $CD4_{NC}$ (Fig. 4e), even though it is the most abundant cell type, where stronger predictive performance would be expected if genetic regulation were present. This cell-type-specific signal in $Mono_{NC}$ is further corroborated by the independent DICE study, where the eQTLs associated with *FLOT1* were

found to be more significant in non-classical monocytes[16]. Third, the association between GReX and disease may be inherently cell-type-specific. While *PSMB9* was significantly predictable in several cell types, its association with RA was observed only in CD8$_{ET}$ cells. For example, although *PSMB9* was more significantly predicted in Mono$_C$ with a *p*-value $6.9 \times 10^{-14}$ (Fig. 4g), as compared to $3.3 \times 10^{-9}$ in CD8$_{ET}$ (Fig. 4f), there was no corresponding signal in the RA GWAS, resulting in a non-significant TWAS association in Mono$_C$. Previous research has reported its cell-type-specific involvement in idiopathic inflammatory myo-pathies in peripheral blood[57], and further investigation into its function in RA is warranted. Lastly, differences in statistical power due to cell type abundance or sample sizes can also lead to apparent cell-type specificity in detecting gene-disease associations. A similar trend of cell-type similarity and specificity was also observed in SLE and asthma (Supplementary Fig. 16), where among the genes identified by scTWAS, 20 of 41 were cell-type-specific for SLE, and 57 of 111 were cell-type-specific for asthma.

To further explore potential cell-type-specific biological pathways revealed by our cell-type-level TWAS, we performed pathway enrich-ment analysis with g:Profiler[58] on significant genes for each disease to test for enrichment of gene ontology (GO) terms. We uncovered sev-eral interesting cell-type-specific pathways. In RA, we identified GO terms related to the production of molecular mediator of immune response exclusively in plasma cells. In an earlier study, plasma cells have been found to express receptor activator of NF-$\kappa$B ligand, a key mediator in osteoclastogenesis[59]. This expression promotes bone resorption and contributes to periarticular bone loss, highlighting the role of plasma cells in both immune response and bone destruction in RA. In SLE, we found GO terms related to extracellular vesicles (EVs) and exosomes in Mono$_{NC}$, which play key roles in intercellular com-munication, carrying inflammatory signals and autoantigens, and thus a crucial physiological and pathological role in SLE[60]. A recent study observed that the Mono$_{NC}$ cells were prominently affected by the exosome, suggesting potential major cell types responsible for exo-some treatment[61]. In asthma, GO terms related to IgE binding and IgE receptor activity were identified in Mono$_C$. IgE is a key mediator in allergic responses, and these findings suggest that Mono$_C$ may play an important role in mediating the allergic inflammatory responses through IgE binding[62,63].

## Cell-subtype-level scTWAS analysis identifies AD-gene associa-tions in microglia subtypes

AD is estimated to affect over 6 million elderly individuals in the United States, and this number is projected to double by 2060[64]. Genetics play an important role in the pathogenesis of AD, with a high heritability of 60%–80%[65]. This motivates large-scale GWAS, which have identified over 70 independent risk loci significantly associated with AD[66]. However, the functional consequences of these AD-associated variants remain to be characterized, especially in the cellular contexts relevant to the pathophysiological processes of AD. Recent advances in highly multiplexed snRNA-seq have facilitated the characterization and identification of these AD-associated cell subtypes in the brain[21,67], such as subpopulations of microglia[21,68–70], astrocytes[21,71,72], and oligodendrocytes[21,72,73]. Gene expression has been found to be het-erogeneous across cell subtypes, and evidence from paired genotype and snRNA-seq data further suggests unique patterns of genetic reg-ulation across cell subtypes[18].

Here, we leveraged scTWAS to investigate the cellular and func-tional mechanisms of AD-associated variants in brain cell subtypes. Using GReX models trained on ROSMAP dorsolateral prefrontal cortex data (Section "scTWAS enhances GReX prediction across cell types in blood and brain tissues"), scTWAS identified a total of 78 genes asso-ciated with AD risk across all cell subtypes, with 33, 18, 20, 21, 22, and 5 genes in subtypes of excitatory neuron, oligodendrocyte, inhibitory neuron, astrocyte, microglia, and OPC, respectively (Supplementary

Data 5). Figure 5 shows that scTWAS reveals the cellular contexts through which AD-associated genes may contribute to disease risk. First, 19 genes were found across multiple cell types (Supplementary Data 5). Notably, *ARL17B* and *KANSL1* were identified in all major cell types, suggesting that the mechanisms involving these genes may operate broadly across cellular contexts. For example, *KANSL1* was previously suggested to be involved in transcription regulation, a fundamental process likely relevant across diverse cell types[74]. In contrast, many scTWAS signals were cell-type-specific (Supplementary Fig. 17). For example, *CR1*, a gene from an established AD locus[75], showed strong associations with AD exclusively in oligodendrocyte subtypes (Fig. 5a and Supplementary Data 5). Prior studies suggest that *CR1* may contribute to the clearance of $\beta$-amyloid (A$\beta$) peptides and modulate immune-related pathways[75,76], with additional evidence for its function in neuronal contexts[77]. These scTWAS findings suggest that *CR1* may play a cell-type-specific role in AD pathogenesis through its activity in oligodendrocytes, motivating additional studies of its reg-ulatory mechanisms in this context. Pathway enrichment analysis fur-ther reveals the enrichment of immune-related biological processes across cell types, the involvement of lipid pathways in astrocyte (reg-ulation of lipid metabolic process) and microglia (protein-lipid com-plex assembly), and pathways identified in only one cell type, such as regulation of catalytic activity and negative regulation of receptor-mediated endocytosis in microglia. We further compared with the bulk-like TWAS evaluated on the same ROSMAP dataset (Section "Prediction model training and evaluation"), which shows that scTWAS identified dozens of associations that were not observed in the bulk-like analysis (Section "Prediction model training and evaluation", Supplementary Fig. 14b).

We next focused on microglia, the brain cell type most enriched for AD genetic risks[19,78], to investigate the potential heterogeneity of AD pathogenesis within this cell population. The ROSMAP dataset annotates 16 subtypes of microglia with distinct biological states, including Mic.13, which is associated with lipid and enriched for sig-natures of disease-associated microglia, and Mic.15, which is enriched for inflammation-related signatures in human[21]. First, several genes were identified as significantly associated with AD across multiple microglia subtypes (Fig. 5b). These include genes located in estab-lished AD GWAS loci, whose functional roles in immune pathways or in microglia are supported by prior literature (e.g., *PICALM*[79], *HLA-DRB1*[80], *EPHA1-AS1*[78], *PTK2B*[81], *BIN1*[82]). In addition, these results included genes such as *DLEU1*, *RASGEF1C* that have not been previously reported in AD GWAS loci[18], warranting future functional studies to investigate their potential roles in AD pathogenesis within microglia. We note that although *APOE* has previously been identified as a AD TWAS gene in microglia[83], we did not highlight it in our scTWAS results because it did not pass Stage 1 screening, despite nominally significant Stage 2 associations in proliferate, surveilling, and reacting microglia (nominal *p*-values < 0.05). Similar results are observed for AN-TWAS and NA-TWAS. This likely reflects the limited sample size, sparsity, and noise of the single-cell dataset, and we discuss plans to further improve GReX prediction in the "Discussion".

Next, we highlight two examples that demonstrate subtype-specific signals captured by scTWAS. First, scTWAS identified *MS4A6A* as significantly associated with AD specifically in Mic.13, whereas its Stage 1 prediction and Stage 2 association tests were generally insig-nificant across other microglial subtypes (Fig. 5b). *MS4A6A* has been previously implicated in AD TWAS[18,84] and is known to play an impor-tant role in microglia[85,86]. Notably, despite the relatively low abundance of Mic.13, the association signal was strong (*p*-value = $1.4 \times 10^{-10}$), suggesting that AD-associated variants may exert strong and subtype-specific regulatory effects on *MS4A6A* in disease-associated microglia, which were not observed in surveilling or reacting microglia.

As a second example, scTWAS identified *PPP1R37* as significantly associated with AD exclusively in Mic.15. *PPP1R37* is located near the

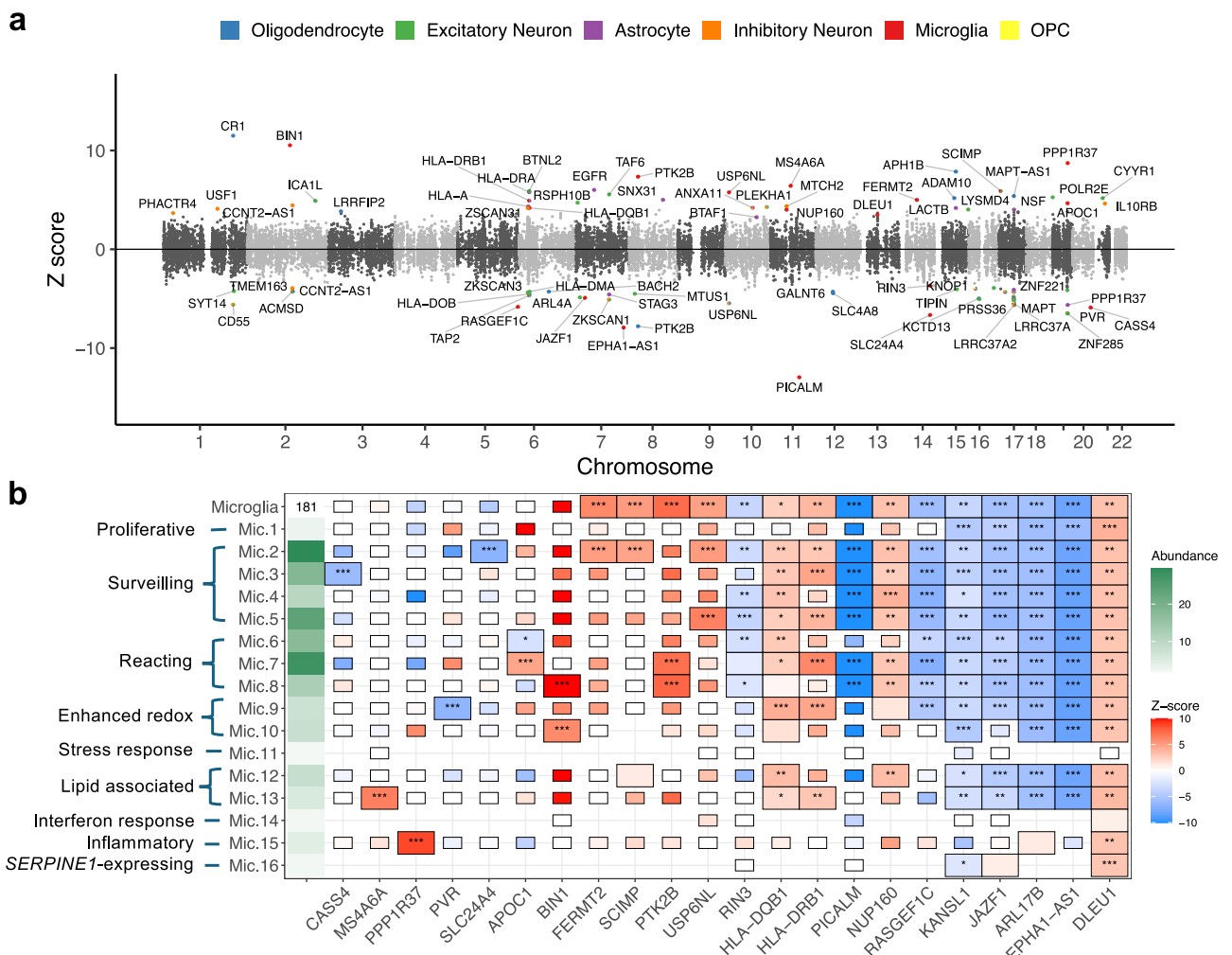

**Fig. 5 | Cell-subtype-level TWAS results of Alzheimer's disease. a** Miami plot of scTWAS results for AD. Each point represents a gene and displays the average Z-score across all cell subtypes in which the gene is significantly associated with AD. The color indicates the major cell type, with significance observed in at least one of its corresponding subtypes. **b** Heatmap of 22 AD-associated genes identified by scTWAS in at least one microglia subtype. The leftmost column shows the abundance of cell (sub)types, calculated as the number of cells/424, and the major cell type of microglia has an abundance of 181. Rectangles denote the results of Stage 1 modeling: a full-size rectangle represents a significant Stage 1 model; a half-size rectangle indicates a non-significant model; and the absence of a rectangle implies no model was available. Asterisks denote Stage 2 association significance: "***" for Bonferroni corrected *p*-value < 0.05, "**" for *p*-value < 0.01, and `*' for *p*-value < 0.05. The two-sided *p*-values are calculated based on TWAS Stage 2 test.

APOE locus, so association signals in this region should be interpreted cautiously given strong LD and nearby AD associations, and the function of *PPP1R37* in brain cell types has not been well characterized. Interestingly, a previous study in a Japanese cohort reported that a *PPP1R37* SNP remained nominally significant for AD risk even after adjusting for the number of APOE *ε*4 alleles[87], suggesting a potentially independent role of this gene in disease susceptibility. A hippocampal TWAS also identified an association between *PPP1R37* and AD[88]. Despite the low abundance of Mic.15, the association signal of *PPP1R37* was highly significant (*p*-value = 2.9 × 10⁻¹⁸), suggesting that AD-associated variants may regulate *PPP1R37* specifically in the inflammatory microglial subtype. Finally, both of these associations, *MS4A6A* in Mic.13 and *PPP1R37* in Mic.15, were identified only by scTWAS and not detected by AN-TWAS or NA-TWAS, highlighting scTWAS's ability to capture biologically meaningful and cell-subtype-specific signals in AD.

## Discussion

In this study, we proposed scTWAS, a novel statistical framework for conducting TWAS using population-scale single-cell data. The method is built on an expression-measurement model[25,31], which quantifies the genetic regulation of underlying gene expression while accounting for noises and technical variations in observed gene counts. To estimate model parameters, we developed an efficient IRLS algorithm that accounts for the heteroskedasticity inherent in single-cell data. Compared with existing TWAS analyses that use single-cell data[28,29], scTWAS avoids the need for heuristic data normalization and instead models the data following a more biologically motivated data-generating mechanism. This allows us to estimate GReX more accurately, leading to improved power in the downstream association testing. Numerical results showcased the improved performance of scTWAS compared to other approaches that rely on data normalization across various cell types, with particularly large gains in those with lower abundance.

We applied scTWAS to two population-scale single-cell datasets to train GReX models and perform gene-based association tests across various traits and diseases. Using data from the OneK1K study, we built cell-type-specific GReX models across 13 immune cell types and conducted TWAS for 29 hematological traits and three autoimmune diseases: RA, SLE, and asthma. We also applied scTWAS

on data from the ROSMAP study to build GReX models across 6 major cell types and 75 cell subtypes in brain tissues, and conducted TWAS for AD. The cell-(sub)type-level TWAS analysis replicated disease-associated genes that were previously identified in tissue-level studies, while providing more granular insights into the cellular contexts in which these associations arise. For example, *IRF5*, a well-established gene associated with RA and SLE, was specifically identified in monocytes and natural killer cells. Importantly, because both the genetic regulation of gene expression and the effect of GReX on disease might differ across cell types, bulk tissue-level analyses may overlook these genes. scTWAS identified novel gene-trait associations that were not identified in previous tissue-level studies, highlighting the necessity of conducting cell-type-level TWAS. Moreover, we also identified cell-type-specific disease-associated genes that appeared uniquely in one specific cell (sub) type. For example, *MS4A6A*, a well-established AD-gene, was identified only in Mic.13, a microglial subtype associated with lipid metabolism and enriched for disease-associated microglia signatures, highlighting its cell-type-specific role and therapeutic relevance. These results show that scTWAS successfully captured known signals and pointed to previously underappreciated cellular mechanisms in disease pathogenesis, which are potentially masked in bulk-level analyses.

A recent study by ref. 89 also introduced a new cell-type-specific TWAS framework called scPrediXcan, which leverages Enformer[90], a deep learning-based sequence-to-function model, to generate in-silico cell-type-specific gene expression for TWAS analysis. Unlike traditional TWAS methods, scPrediXcan does not use paired genotype and gene expression data across individuals to estimate genetic effects. Instead, it leverages Enformer-derived epigenomic features to generate in-silico cell-type-specific gene expression from DNA sequence alone, and then performs TWAS using these in-silico expression profiles. While scPrediXcan offers an exciting opportunity to extend TWAS to cell types lacking population-level single-cell data, performing TWAS using in-silico expression introduces important considerations. Because the predicted in-silico expression reflects regulatory relationships imputed by a deep learning model, rather than the observed genotype-driven variation in gene expression across individuals, the resulting associations may capture model-driven regulatory patterns. Recent studies have shown that these patterns may have limited consistency with observed genotype-expression associations across individuals[91,92]. In contrast, scTWAS directly uses paired genotype and scRNA-seq count data and a statistically principled framework to learn genotype-driven variation in gene expression across individuals. Thus, the two methods are designed for different applications. When population-scale single-cell data are not available for the cell type of interest, scPrediXcan provides sequence-based regulatory predictions, though its interpretation depends on the deep learning model's accuracy in predicting epigenomic features across individuals. scTWAS directly leverages population-scale single-cell data, which are increasingly available for diverse tissues and diseases[24,93,94], to learn the observed association between genotype and gene expression in cell types.

While scTWAS provides a powerful statistical framework for cell-type-specific TWAS, several limitations should be noted. First, although the method can be conceptually viewed as a two-stage least squares approach, the resulting associations are not necessarily causal. Due to LD, multiple genes in the same region may appear associated, and additional challenges such as horizontal pleiotropy or shared regulatory variants across genes and cell types can complicate interpretation[95]. A natural extension of scTWAS is to incorporate fine-mapping strategies[96–98] that more precisely distinguish true causal genes from LD-linked neighbors. In addition, recent approaches that fine-map gene-tissue pairs[99,100] can be extended to the single-cell setting to provide evidence about whether a gene's effect is specific to a particular cell type or shared across multiple cell types. Integrating such cell-type-aware fine-mapping with scTWAS would further enhance the resolution and interpretability of scTWAS signals and represents an important direction for future methodological development. Second, in its current form, scTWAS trains a separate GReX prediction model for each cell type independently. While this design captures cell-type-specific genetic effects, it does not borrow strength across cell types, which could be valuable given the shared genetic regulation of gene expression in related cell populations[27]. A promising future direction is to extend the framework to jointly model multiple cell types, with the goal of improving prediction accuracy by sharing information across cell types while still preserving cell-type-specific signals. Third, scTWAS treats all SNPs equally when building the prediction model. However, incorporating functional annotations to prioritize SNPs with regulatory potential, such as chromatin accessibility or transcription factor binding, could further enhance prediction performance as well as model interpretability. This strategy has shown promise in related methods[69], and we leave this as an important direction for future work. Finally, scTWAS currently operates within a single-ancestry framework. Extending the method to multi-ancestry settings[101,102] would increase applicability and improve fine-mapping resolution, as differences in LD structure and ancestry-specific genetic effects can provide additional power and resolution beyond single-population analyses. Incorporating these considerations represents a promising avenue for future work.

In summary, we introduce scTWAS, a powerful and biologically grounded statistical framework for performing TWAS using population-scale single-cell data. As these data become increasingly available[24,93,94], scTWAS provides a promising tool for building cell-type-specific GReX models and identifying cell-type-specific gene-trait associations for more diseases in disease-relevant cell types. We anticipate that scTWAS will facilitate novel biological insights, refine the mapping from genotype to phenotype, and ultimately contribute to a deeper understanding of disease mechanisms at the cellular level.

## Methods

### scTWAS method

scTWAS adopts a moment-based estimation framework to estimate $\boldsymbol{\beta}$ for GReX prediction. For a given pair of gene and cell type, let $x_i$ denote the observed pseudo-bulk counts for the gene in the cell type of interest, $s_i$ the sequencing depth. We use $\mathbf{g}_i$ to denote the *cis*-genotypes for individual $i$, and $\mathbf{c}_i$ the vector of covariates. The expected underlying expression level for individual $i$ is given by $\mu_i = \mathbf{g}_i'\boldsymbol{\beta} + \mathbf{c}_i'\boldsymbol{\gamma}$, and $\sigma_i^2$ denotes the biological variance.

Under the expression-measurement model Eqs. (1) and (2), the observed pseudo-bulk counts $x_i$ satisfy the following moment conditions:

$$\mathrm{E}(x_i) = s_i\mu_i = s_i(\mathbf{g}_i'\boldsymbol{\beta} + \mathbf{c}_i'\boldsymbol{\gamma}), \tag{3}$$

$$\mathrm{Var}(x_i) = s_i\mu_i + s_i^2\sigma_i^2, \tag{4}$$

This motivates our scTWAS Stage 1 model:

$$x_i = s_i \cdot (\mathbf{g}_i'\boldsymbol{\beta} + \mathbf{c}_i'\boldsymbol{\gamma}) + \epsilon_i = s_i\mathbf{g}_i'\boldsymbol{\beta} + s_i\mathbf{c}_i'\boldsymbol{\gamma} + \epsilon_i, \tag{5}$$

where $\epsilon_i$'s are independent mean-zero random errors. The variance of $\epsilon_i$ is given by Eq. (4), which may vary greatly across samples due to technical variations in sequencing depths. This leads to heteroskedasticity and statistically inefficient estimates if left unaddressed. To mitigate this, we propose to estimate $\boldsymbol{\beta}$ by minimizing the following

weighted least squares loss function with an elastic net penalty on $\boldsymbol{\beta}$:

$$L(\boldsymbol{\beta}, \boldsymbol{\gamma}) = \sum_{i=1}^{n} \omega_i (x_i - s_i \mathbf{g}_i' \boldsymbol{\beta} - s_i \mathbf{c}_i' \boldsymbol{\gamma})^2 + \lambda(0.5||\boldsymbol{\beta}||_1 + 0.5||\boldsymbol{\beta}||_2^2), \quad (6)$$

where we use $\omega_i = 1/\text{Var}(x_i)$ as weights to downweight observations with higher variance and upweight those with lower variance, thereby enhancing statistical efficiency. We implement the optimization of Eq. (6) with R package `glmnet`[103], where the response is defined as $x_i\sqrt{\omega_i}$, the design matrix consists of genotypes $\mathbf{g}_i$ and covariates $\mathbf{c}_i$ scaled by sequencing depth $s_i$ and $\sqrt{\omega_i}$, and no penalty is imposed on $\boldsymbol{\gamma}$. We further use cross-validation to select the hyperparameter $\lambda$ which minimizes the out-of-sample loss function.

As shown in Eqs. (4) and (3), the weights $\omega_i$'s used in Eq. (6) depend on unknown parameters $\boldsymbol{\beta}$ (and $\boldsymbol{\gamma}$). To address this, we propose an IRLS algorithm that iterates between updating $\omega_i$'s and updating $\boldsymbol{\beta}$, $\boldsymbol{\gamma}$. In specific, we update $\boldsymbol{\beta}$, $\boldsymbol{\gamma}$ given $\omega_i$'s by optimizing Eq. (6), and update $\omega_i$'s given $\boldsymbol{\beta}$, $\boldsymbol{\gamma}$ by setting $\sigma_i^2 = \mu_i^2/\theta$ and leveraging Eqs. (1) and (4). Here $\theta$ denotes the over-dispersion parameter, encoding a mean-variance dependency as commonly observed in bulk and single-cell RNA-seq data[26,32]. We estimate $\theta$ and the initial weights using R package `sctransform`[26] under a null model with no genetic effects. The IRLS algorithm is presented in Box 1, which typically converges within ten iterations. See Supplementary Methods for more details on the algorithm.

For genes that are well-predicted (as evaluated by $p$-values of prediction in fivefold cross-validation[4,5]), Stage 2 of scTWAS tests for the association between predicted gene expression and the trait of interest. When only the GWAS summary statistics are available, the association between predicted expression and the trait can be expressed as[5]:

$$\mathbf{z}_{\text{TWAS}} = \frac{\widehat{\boldsymbol{\beta}}' \mathbf{z}_{\text{GWAS}}}{\sqrt{\widehat{\boldsymbol{\beta}}' \mathbf{V} \widehat{\boldsymbol{\beta}}}}, \quad (7)$$

where $\widehat{\boldsymbol{\beta}}$ is the estimated SNP weights in Stage 1, $\mathbf{z}_{\text{GWAS}}$ is a vector of GWAS summary $Z$-scores of the corresponding $cis$-SNPs, and $\mathbf{V}$ is the LD correlation matrix of the $cis$-SNPs, which can be estimated by a reference panel. Under the null hypothesis of no association, $\mathbf{z}_{\text{TWAS}}$ follows a standard normal distribution and the $p$-value can be calculated analytically.

---

**Box 1: IRLS for building Stage 1 models in scTWAS.**

**Require:** Pseudo-bulk counts of a gene from $n$ individuals $X = (x_i)_{n \times 1}$ in the cell type of interest, standardized genotype matrix of $cis$-SNPs around the gene $\mathbf{G} = [\mathbf{g}_1', \ldots, \mathbf{g}_n']'$, covariate matrix (including intercept) $\mathbf{C} = [\mathbf{c}_1', \ldots, \mathbf{c}_n']'$, sequencing depths $\mathbf{s} = [s_1, \ldots, s_n]'$, initial weights $\boldsymbol{\omega}^{(0)} = [\omega_1^{(0)}, \ldots, \omega_n^{(0)}]'$, overdispersion parameter $\theta$, number of iterations $m > 0$

1: **for** $t = 1, \ldots, m$ **do**
2: $\mathbf{U}^{(t)} \leftarrow [\mathbf{G} \odot \mathbf{s} \odot \sqrt{\boldsymbol{\omega}^{(t-1)}}, \mathbf{C} \odot \mathbf{s} \odot \sqrt{\boldsymbol{\omega}^{(t-1)}}], \mathbf{y}^{(t)} \leftarrow \mathbf{x} \odot \sqrt{\boldsymbol{\omega}^{(t-1)}}$
3: ▷ $\odot$ *denotes the element-wise product (by row).* ◁
4: $\boldsymbol{\beta}^{(t)}, \boldsymbol{\gamma}^{(t)} \leftarrow$ `cv.glmnet`$(\mathbf{y} = \mathbf{y}^{(t)}, \mathbf{x} = \mathbf{U}^{(t)}, \ldots)$
5: ▷ *`cv.glmnet` is a function from R package `glmnet`. See Supplementary Methods for details on how to set other arguments.* ◁
6: $\mu_i^{(t)} \leftarrow \mathbf{g}_i' \boldsymbol{\beta}^{(t)} + \mathbf{c}_i' \boldsymbol{\gamma}^{(t)}, i = 1, \ldots, n$
7: $\omega_i^{(t)} \leftarrow 1/(s_i \mu_i^{(t)} + s_i^2 (\mu_i^{(t)})^2/\theta), i = 1, \ldots, n$
8: $\widehat{\boldsymbol{\beta}} = \boldsymbol{\beta}^{(m)}$
9: **return** $\widehat{\boldsymbol{\beta}}$

---

## Other TWAS methods under comparison

In this work, we compare scTWAS with two existing approaches that use single-cell data for TWAS analysis, referred to as NA-TWAS[29], which applies Normalization followed by Aggregation, and AN-TWAS[28], which applies Aggregation followed by Normalization. Both approaches rely on the traditional TWAS framework[4,5] developed for bulk gene expression data, which assumes the following Stage 1 model on normalized gene expression data $\widetilde{x}_i$:

$$\widetilde{x}_i = \mathbf{g}_i' \boldsymbol{\beta} + \mathbf{c}_i' \boldsymbol{\gamma} + \widetilde{\epsilon}_i, \quad (8)$$

where $\widetilde{\epsilon}_i$ denotes random noise. Throughout, we used elastic net penalty to estimate $\boldsymbol{\beta}$, $\boldsymbol{\gamma}$ by minimizing the following loss function

$$\sum_{i=1}^{n} (\widetilde{x}_i - \mathbf{g}_i' \boldsymbol{\beta} - \mathbf{c}_i' \boldsymbol{\gamma})^2 + \lambda(0.5||\boldsymbol{\beta}||_1 + 0.5||\boldsymbol{\beta}||_2^2), \quad (9)$$

while other penalty functions on $\boldsymbol{\beta}$ can also be used. This framework has been widely applied to normalized and continuous bulk gene expression traits with relatively homoskedastic variance[4,5].

To be compatible with the above traditional TWAS framework, NA-TWAS and AN-TWAS propose to apply various normalization strategies prior to applying the framework (Supplementary Fig. 1). In specific, NA-TWAS begins by applying counts per million normalization to the cell-by-gene count matrix for a specific cell type, followed by log transformation and mean aggregation across cells for each individual; AN-TWAS first aggregates single-cell counts by individual to create a pseudo-bulk count matrix, then applies TMM normalization (via the R package `edgeR`) and log transformation. Finally, both approaches perform inverse normal transformation.

## Data sets

### Genotype and scRNA-seq data

**The OneK1K study.** The OneK1K study comprises scRNA-seq data from 1.27 million PMBCs collected from 982 donors[17]. Cells have been classified into 14 immune cell types by the authors, including plasma cells (Plasma); dendritic cells (DC); two B cell types—immature and naïve B ($B_{\text{IN}}$) cells and memory B ($B_{\text{Mem}}$) cells; two natural killer cell types—natural killer (NK) cells and NK recruiting ($NK_R$) cells; two monocyte cell types—classical ($\text{Mono}_C$) and non-classical ($\text{Mono}_{NC}$) monocytes; three CD4$^+$ T cell types—CD4 naïve and central memory T cells ($\text{CD4}_{NC}$), CD4$^+$ T cells with an effector memory or central memory phenotype ($\text{CD4}_{ET}$), and CD4$^+$ T cells expressing SOX4 ($\text{CD4}_{\text{SOX4}}$); and three CD8$^+$ T cell types—CD8 naïve and central memory T cells ($\text{CD8}_{NC}$), CD8$^+$ T cells with expression of S100B ($\text{CD8}_{\text{S100B}}$), CD8$^+$ T cells with an effector memory phenotype ($\text{CD8}_{ET}$)[17]. For each cell type, we constructed a pseudo-bulk count matrix by extracting unique molecular identifier (UMI) counts across all cells and aggregating them for each individual. Genes with pseudo-bulk UMI counts less than 1000 were filtered out. Finally, $\text{CD4}_{\text{SOX4}}$ was excluded from the analysis due to the limited number of genes remained. The number of samples, number of cells, and number of genes retained in the analysis for each cell type is provided in Supplementary Data 1. The genotype data were obtained from the authors, where 483,255 autosomal SNPs were genotyped and imputed to the Haplotype Reference Consortium panel, with only SNPs having an imputation $R^2 > 0.8$ being retained. We further performed quality control using PLINK 2.0[104] to retain variants with minor allele frequency $> 0.01$ and in Hardy–Weinberg equilibrium (`-maf 0.01 -hwe 1e-6`).

**The ROSMAP study.** The ROSMAP study is designed to study aging and dementia with multi-omics profiling of human brain, including whole genome sequencing (WGS) data[34] and population-scale single-nucleus RNA-sequencing data on postmortem brain samples[18,35]. We performed similar quality control of genotype data using PLINK 2.0. To train Stage 1 prediction models, we used the snRNA-seq data on dorsolateral prefrontal cortex from[18], which profiled 1.65 million nuclei and identified a total of 16 major cell types and 95 cell subtypes. Among 437 individuals with snRNA-seq data, 424 have both genotype and snRNA-seq data available. We excluded major cell types with less

than 10,000 cells and focused on the remaining six major cell types (excitatory neurons, oligodendrocytes, inhibitory neurons, astrocytes, microglia, and OPCs) and their 75 subtypes. For example, there are 16 subtypes of microglia under various biological states in this dataset (Fig. 5b). For each of these cell (sub)types, we constructed pseudo-bulk count matrices similar to the procedure above and filtered out genes with less than 1000 total counts. The number of samples, number of cells, and number of genes retained in the analysis for each cell (sub)type are provided in Supplementary Data 1. The only exception is the analysis of microglia subtypes in Fig. 5, where we focused on the genes with at least 1000 total counts in the microglia major cell type to expand the set of genes studied within the cellular context critical to AD mechanisms. For cross-study validation, we further used an independently generated snRNA-seq dataset on prefrontal cortex[35], that profiled 2.3 million nuclei and identified seven major cell types. Among 427 individuals with snRNA-seq data, 396 have both genotype and snRNA-seq data available. We focused on excitatory neurons, oligodendrocytes, inhibitory neurons, astrocytes, microglia, and OPCs to validate the GReX models trained on data from dorsolateral prefrontal cortex[18].

**The DICE project.** The DICE project collects genotype data and bulk RNA-sequencing data on FACS-sorted immune cells from 91 subjects[16]. We used the genotype data from the original publication, which have been imputed with the 1000 Genomes Project phase 3[105]. We used the bulk gene expression data from the original publication, which have been normalized with Transcript Per Million (TPM)[106]. Among the 15 immune cell types profiled in DICE, we matched six of them to the most similar cell types in OneK1K, including classical Monocytes, non-classical Monocytes, CD16 NK cells, Naive B cells, Naive CD4 T cells (matched to $CD4_{NC}$ in OneK1K), and Naive CD8 T cells (matched to $CD8_{NC}$ in OneK1K).

**GWAS summary data.** GWAS summary statistics for hematological traits were generated by Neale Lab (https://www.nealelab.is/uk-biobank) using UK Biobank (UKB) White ancestry data ($N = 349,856$). GWAS summary statistics for the three immune-related diseases (RA, SLE, and Asthma) were obtained from[46] ($N_{cases} = 14,361, N_{controls} = 43,923$)[47], ($N_{cases} = 5,201, N_{controls} = 9,066$)[48], ($N_{cases} = 88,486, N_{controls} = 447,859$), respectively. These GWAS summary statistics were then imputed using ImpG[107] with OneK1K samples as the reference panel. GWAS summary statistics for AD were obtained from[66], which were imputed using ImpG with ROSMAP WGS data as the reference panel. We further conducted quality control on the imputed GWAS data, including filtering out imputed SNPs with imputation accuracy $R^2 < 0.6$, and any SNPs with ambiguous alternative alleles.

**Simulation settings**
**One-sample individual-level TWAS.** We performed simulation studies to evaluate scTWAS with genotype and gene expression data of 982 individuals from the OneK1K study. We simulated the raw count $x_{ij}$ for the $j$-th cell of the $i$-th individual with a Poisson-Gamma model, and the trait of interest $y_i$ of the $i$-th individual as follows:

$$\mu_i = \mathbf{g}_i'\boldsymbol{\beta} + \beta_0, \quad z_{ij} \sim \text{Gamma (shape} = \theta, \text{scale} = \mu_i/\theta); \quad (10)$$

$$x_{ij}|z_{ij} \sim \text{Poisson}(s_{ij}(z_{ij} + a_i)), \text{ where } \alpha_i \sim \mathcal{N}(0, \beta_0/100); \quad (11)$$

$$y_i = \mu_i + e_i, \text{ where } e_i \sim \mathcal{N}\left(0, \frac{1 - h_g^2}{h_g^2}\text{Var}(\boldsymbol{\mu})\right), h_g^2 = 0.05. \quad (12)$$

In Eq. (10), $\mathbf{g}_i$ were the standardized genotypes of the *cis*-SNPs around a gene, and 10 causal *cis*-SNPs were randomly selected with the corresponding weights generated from Uniform $((-3 \times 10^{-6}, -5 \times 10^{-9}) \cup (5 \times 10^{-9}, 3 \times 10^{-6}))$. The intercept $\beta_0$ was set to $1 \times 10^{-4}$, and $\theta$ was set to 10. In Eq. (11), the sequencing depth of the $j$-th cell $s_{ij}$ in a specific cell type was directly extracted from the OneK1K real data. A random intercept $\alpha_i$ was simulated to introduce correlations among cells within the $i$-th individual[108]. In Eq. (12), the GWAS trait $y$ was simulated such that the genetically regulated gene expression accounted for around 5% of its variance ($h_g^2 = 0.05$). We used this relatively large per-gene trait heritability $h_g^2$ to ensure nontrivial Stage 2 association power given the small GWAS sample size ($N_{GWAS} = 982$) in one-sample simulations.

We also considered two scenarios to evaluate the type I-error rate of the methods. In the first setting, we simulated the expected underlying gene expression level $\mu_i = \beta_0$ independent of the genotype data (*cis*-SNPs). In the second setting, we simulated the GWAS trait from a standard normal distribution independent of the gene expression.

We selected six cell types ($CD4_{NC}$, $CD8_{ET}$, $B_{IN}$, $Mono_C$, $NK_R$, and Plasma) for the simulation, representing different levels of cell type abundance in the real data. Twenty genes were randomly selected and 50 replicates were performed for each gene-cell type pair, with a total of 1000 simulation replicates in a specific cell type. In each replicate, we only replaced the count of the specific gene with the simulated single-cell count, while keeping the counts of other genes the same as the OneK1K real data.

With the cell by gene count matrix, we applied the proposed scTWAS, and NA-TWAS and AN-TWAS described in Section "Other TWAS methods under comparison". To evaluate the power of TWAS analysis, we defined a significant gene-trait association as one where the prediction $p$-value < 0.05 in fivefold cross-validation in Stage 1 (see Section "Prediction model training and evaluation"), and the Stage 2 association was also significant with a $p$-value < 0.05/$M$, where $M$ corresponded to the number of tests in the real data application in each cell type. To evaluate the type-I error, we defined a significant gene-trait association as one where both the Stage 1 GReX prediction in cross-validation and the Stage 2 association were significant (nominal $p$-value < 0.05).

**Two-sample TWAS with GWAS summary data.** We extended our simulation design to a two-sample framework. Stage 1 single-cell gene expression data were generated the same way as in Section "One-sample individual-level TWAS" using OneK1K individual-level data, and for Stage 2, GWAS summary data for the complex trait were generated directly.

For a given pair of gene and cell type, we computed the per-SNP genotype standard deviations $D_{jj} = \sqrt{2p_j(1 - p_j)}$, where $p_j$ is the minor allele frequency, and the LD matrix $\mathbf{V}$ from the OneK1K dataset. Given the true (joint) SNP effect on the GWAS trait $\mathbf{b}_{GWAS}$, the observed marginal GWAS summary statistics were generated as follows[109]:

$$\hat{\boldsymbol{\beta}}_{GWAS} \sim \mathcal{N}\left(\mathbf{D}^{-1}\mathbf{VD}\,\mathbf{b}_{GWAS}, (\sigma^2/N_{GWAS})\mathbf{D}^{-1}\mathbf{VD}^{-1}\right), \text{SE}_j$$
$$= \frac{\sigma}{\sqrt{N_{GWAS}\,2p_j(1 - p_j)}}, \quad (13)$$

where $\mathbf{D} = \text{diag}(D_{jj})$, $N_{GWAS}$ is the sample size of GWAS, and $\sigma$ is the residual standard deviation of the GWAS trait. To generate the joint SNP effect on the GWAS trait $\mathbf{b}_{GWAS}$, we set $\mathbf{b}_{GWAS} = \gamma\boldsymbol{\beta}$, where $\boldsymbol{\beta}$ is the SNP effect on the gene expression generated in Stage 1. We then chose $\gamma = \sqrt{h_g^2/\text{Var}(\mathbf{G}\boldsymbol{\beta})}$, where $\text{Var}(\mathbf{G}\boldsymbol{\beta}) = \boldsymbol{\beta}'\mathbf{DVD}\boldsymbol{\beta}$, so that the trait heritability explained by the genetically regulated gene expression is $h_g^2$, and $\sigma^2 = 1 - h_g^2$.

In this simulation, we focused on gene *AAK1* on chromosome 2, and evaluated three immune cell types, CD8$_{ET}$ (abundant), Mono$_C$ (less abundant), and NK$_R$ (rare). Stage 1 GReX prediction weights were trained from individual-level data using either a subsample of $N_{eQTL}$ = 500 or all 982 OneK1K individuals. Stage 2 GWAS summary statistics were generated at sample sizes $N_{GWAS} \in \{3 \times 10^4, 5 \times 10^4, 10^5\}$ under the LD-aware model described above. We considered *cis* architectures in which either 10 or 3 causal *cis*-SNPs had nonzero effects in **β**. We scaled the effect size to achieve per-gene trait heritability $h_g^2 \in \{0, 0.05/180, 0.10/180, 0.20/180\}$, where $h_g^2 = 0$ represents type-I error (null) settings. For each configuration, 100 simulation replicates were performed.

To assess robustness, we examined an out-of-sample LD setting by resampling an independent LD matrix around the OneK1K LD. We also considered horizontal pleiotropy by adding an independent direct effect of *cis*-SNPs. Full details are provided in the Supplementary Section 1.3.

### Real data applications

**Prediction model training and evaluation.** With the aforementioned genotype and scRNA-seq data in Section "Genotype and scRNA-seq data", we built Stage 1 prediction models for each gene-cell type pair. For each gene, *cis*-SNPs within 500 kB upstream of the gene transcription start site and 500 kB downstream of the transcription end site were used as predictors. Additionally, to ensure a high overlap between the SNPs in the prediction models and those in the GWAS summary statistics, we only considered SNPs present in the UKB GWAS summary statistics for the OneK1K models, and those present in the AD GWAS[66] for the ROSMAP models. For OneK1K models, we adjusted for covariates including sex, age, and six genotype principal components provided by the authors. For ROSMAP models, we adjusted for covariates including the first three genotype PCs, age at death, sex, post-mortem interval, study (ROS or MAP), and a hidden batch effect covariate inferred by principal component analysis (Supplementary Methods, Supplementary Figs. 19 and 20). R package `glmnet` was used to train the elastic net model for all three methods with the mixing value $\alpha = 0.5$ and the tuning parameter $\lambda$ was selected using fivefold cross-validation.

We evaluated the within-study prediction accuracy by fivefold cross-validation. For NA-TWAS and AN-TWAS, we regressed the observed normalized expression on the predicted expression across all predicted folds as implemented in FUSION-TWAS. For scTWAS, we used weighted linear regression to regress the observed pseudo-bulk counts on the predicted $s \odot G\widehat{\boldsymbol{\beta}}$ with estimated weights across all predicted folds. Only genes significantly predicted in Stage 1, defined as those whose estimated GReX is predictive of observed gene expression in cross-validation (Bonferroni corrected *p*-value < 0.05), were included in Stage 2 analysis to test for the association with the trait of interest.

We also evaluated the cross-study prediction performance of three cell-type-specific TWAS methods. For the GReX models of immune cell types trained with OneK1K data, we used the external validation dataset from the DICE project[16]. We focused on 6 cell types that match between OneK1K and DICE, including classical monocytes, non-classical monocytes, NK cells, naive B cells, naive CD4$^+$ T cells, and naive CD8$^+$ T cells. For each cell type and method considered, we used the prediction model trained with the corresponding cell type from OneK1K data and the DICE genotype data to predict cell-type-specific gene expression for DICE subjects. We calculated the Pearson correlation between predicted gene expression and log(TPM+1) normalized bulk gene expression data from DICE, and compared the number of significantly predicted genes (nominal *p*-value < 0.05, Pearson's correlation) across three methods. For the GReX models of brain cell types trained with ROSMAP data on dorsolateral prefrontal cortex[18], we used an independently generated snRNA-seq dataset on prefrontal

cortex from ref. 35 for external validation. We focused on six cell types that overlap between two studies (excitatory neuron, oligodendrocyte, inhibitory neuron, astrocyte, microglia, and OPC) and used the prediction models trained with the corresponding cell type in ref. 18 and the genotype data to predict cell-type-specific gene expression in ref. 35. The prediction performance of each method is evaluated by prediction *p*-values, as defined above, following the approach used in cross-validation.

In addition to the three cell-type-specific methods, we also constructed a bulk-like TWAS model for comparison. Specifically, we aggregated cells across all cell types in the single-cell dataset to mimic bulk expression in tissue samples, which enables a controlled comparison using the same single-cell dataset. We then applied the same normalization procedure used in AN-TWAS (see Supplementary Fig. 1). Using the synthetic bulk expression data, we trained bulk GReX prediction models and evaluated their cross-validation prediction accuracy.

**TWAS analysis using GWAS summary statistics.** We applied cell-type-level TWAS to GWAS summary data using FUSION-TWAS[5,110]. FUSION-TWAS leverages GWAS summary statistics, a gene expression predictive model, and an LD reference panel to test the association between genetically predicted gene expression and a phenotype of interest, analyzing one gene at a time. Specifically, we conducted TWAS analysis on 29 hematological traits and 3 immune-related diseases using Stage 1 models trained on OneK1K immune cell types with scTWAS, NA-TWAS, and AN-TWAS, respectively, and the OneK1K study as the LD reference panel. We also conducted TWAS analysis on AD using Stage 1 models trained on ROSMAP brain cell (sub)types with scTWAS, NA-TWAS, and AN-TWAS, respectively, and the ROSMAP study as the LD reference panel. To account for the large number of hypotheses tested, we used Bonferroni corrected *p*-value threshold of 0.05/*M* to identify significant gene-trait associations in a specific cell type of interest, where *M* was the number of genes significant in at least one prediction model. For comparison with the bulk-like TWAS analysis, we also conducted TWAS using the GReX prediction models trained on aggregated single-cell data.

To define a locus, we started by ranking significant genes according to their TWAS *p*-values, with higher ranks assigned to smaller *p*-values. For the top-ranked gene, we defined a locus as a 1 million base pair window around it and included all genes within this region as part of the locus. This process was repeated for the remaining significant genes until all significant genes were associated with loci.

### GO enrichment analysis

In Section "Real data applications", we used `gost` in R package `gprofiler2` (v0.2.2) to perform enrichment analysis for GO with one-sided Fisher exact tests and selected GO terms with adjusted *p*-value < 0.05 using the default "`g_SCS`" method.

### Reporting summary

Further information on research design is available in the Nature Portfolio Reporting Summary linked to this article.

## Data availability

The GReX prediction models for immune and brain cell types generated in this study have been deposited in Figshare at [https://doi.org/10.6084/m9.figshare.31329184] and [https://doi.org/10.6084/m9.figshare.31329139][111,112]. The availability for datasets used to train or validate GReX models is as follows: OneK1K genotype and scRNA-seq data are publicly available from the OneK1K study[17]. Genotype and gene expression data of immune cell types from the DICE project[16] are available under controlled access in dbGap under accession code phs001703.v5.p1 [https://www.ncbi.nlm.nih.gov/projects/gap/cgi-bin/study.cgi?study_id=phs001703.v5.p1], access can be obtained by

requesting Authorized Access on dbGap. WGS and snRNA-seq data from the ROSMAP study are available on Synapse under controlled access with accession codes syn11724057 [https://www.synapse.org/Synapse:syn11724057], syn53366818 [https://www.synapse.org/Synapse:syn53366818], and syn52293417 [https://www.synapse.org/Synapse:syn52293417], access can be obtained by submitting a Data Use Certificate to Synapse.

## Code availability

R package for scTWAS is available at https://github.com/ZhaotongL/scTWAS[113]. Code for real data analyses is available at https://github.com/ZhaotongL/scTWAS_paper[114].

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

## Acknowledgements

Z.L. was supported in part by the CTSA Provost Award at Florida State University. C.S. was supported in part by the URC Research Award at Emory University. We thank Joseph Powell and Angli Xue for help with obtaining the OneK1K data. This work was supported in part by the Rollins School of Public Health High Performance Computing Cluster at Emory University. The results published here are in part based on data obtained from the AD Knowledge Portal.

## Author contributions

C.S. and Z.L. designed the research and wrote the paper; Z.L. implemented the method, performed simulation studies, and analyzed the OneK1K data; C.S. analyzed the ROSMAP data.

## Competing interests

The authors declare no competing interests.
