## [Transparent Peer Review file · Nature Communications]

scTWAS: A powerful statistical framework for single-cell transcriptome-wide association studies

Corresponding Author: Dr Chang Su

Version 0:

Reviewer comments:

Reviewer #1

(Remarks to the Author)

The authors developed a single-cell transcriptome-wide association study (scTWAS) method based on an expression-measurement model to predict cell-type-specific (CTS) genetically regulated gene expression (GReX) and test its association with phenotype. They assume the pseudobulk RNA read counts follow a Poisson distribution with the true underlying expression as a latent variable, which is modeled as a function of genotype without assuming a known distribution. Then, a moment-based method is used for estimation. The authors conducted extensive simulations and real data applications to show that scTWAS outperforms two existing methods. The paper is well written, and the method is a timely contribution to the field. Below, I list some comments that may further improve the manuscript.

The proposed scTWAS and the two existing methods seem to have two differences in GReX prediction: 1) data normalization and transformation, and 2) the variance weighted regression in Eqn 6. My impression is that the main improvement comes from the first component (expression-measurement model). scTWAS needs the variance weighting since it directly models the read counts and may be impacted by skewed values. Can the authors comment on this?

There is a competing method, scPrediXcan, first posted on bioRxiv in 2024 and published recently. The authors may discuss or compare scTWAS's performance with the deep-learning-based scPrediXcan.

The bulk TWAS is known to have a low out-of-sample R^2 for GReX prediction. Is the prediction accuracy improved for CTS TWAS?

APOE is the most established AD risk gene. For CTS-TWAS of AD using GWAS (Jansen et al., Nature Genetics 2019) and snRNA-seq-based eQTL summary statistics (Bryois et al., Nature Neuroscience 2022) with FUSION, APOE is significant in microglia after Bonferroni adjustment. The authors may comment on why APOE is not identified here.

(Remarks on code availability)

Reviewer #2

(Remarks to the Author)

Summary:

Lin and Su proposed a novel method, scTWAS, to extend the traditional TWAS framework to scRNA-seq settings. I found the method interesting and applaud the authors for conducting extensive real data applications. My major concern is whether the simulations reflect real-world scenarios. I've listed my major and minor comments below:

Major Comments

1. In the scTWAS scenario, I think the most interesting question is: given a trait, are the scTWAS-identified genes more cell-type-specific or more shared across cell types? Based on the real data results, can the authors comment on this question or try to quantify it?
2. In the introduction, the authors mentioned both cell types and cell subtypes. I think it's necessary to explicitly define what

is meant by these terms. As a result, I found the statement “the heterogeneity across cell subtypes will be obscured” a bit confusing. That being said, can the authors also clarify how their method disentangles heterogeneity across cell subtypes? Please correct me if I’ve misunderstood anything here.

3. In the Methods section, it’s not clear to me how the genotype data were simulated. Did the simulation reflect the real LD structure across SNPs within a gene window? The manuscript mentions that “10 causal cis-SNPs were randomly selected.” Ten seems like a large number, since most genes only have 1–3 cis-eQTLs. While regularized prediction models often select more SNPs with non-zero weights (e.g., around 10), that reflects predictive modeling, not the true number of causal eQTLs. Strictly speaking, TWAS tests the association between predicted gene expression and the trait—not the actual expression levels—based on the current two-step TWAS framework.

4. In Section 4.4, several parameters and symbols are not defined, so I’m not 100% sure I understood it correctly. Does y represent the complex trait? Does h^2_g represent the trait heritability explained by a single gene? If so, setting $h^2_g = 0.05$ seems very high. For example, if trait heritability is 0.3 and there are 1000 causal genes, per-gene heritability would be $0.3 / 1000 = 3e-4$.

5. In the Methods section, it’s not clear how the GWAS data were simulated. Did the authors account for LD? How was GWAS sample size incorporated? Also, in practice, eQTL and GWAS samples are from different cohorts. Was this reflected in the simulation design?

6. I think some additional simulations are needed, such as: Varying per-gene heritability with more realistic values, Simulating both in-sample and out-of-sample LD, Considering pleiotropic effects of cis-eQTLs, Varying eQTL and GWAS sample sizes, Model misspecification cases, such as when the selected cell type doesn’t match the cell type relevant to the trait.

7. In the real data analysis, especially in visualizations and comparisons, I think the authors should always include conventional bulk RNA-seq TWAS as a baseline. This would give readers a better sense of how much is gained by incorporating scRNA-seq data.

8. In the discussion of limitations and future directions, the authors may want to mention the potential extension to fine-mapping and multi-ancestry settings, along with relevant references. Also, Wainberg et al., Nature Genetics, 2019 is an important citation when discussing limitations of TWAS methods.

Minor Comments

1. In the abstract, the use of the word “overlooking” seems subjective. I don’t think it’s fair to say that TWAS overlooked cell types; rather, the necessary data simply weren’t available when TWAS was first developed.

2. It would be helpful if the authors could add line numbers to the manuscript so reviewers can reference specific sections more easily.

3. Recently, a similar paper by Zhou Y. et al., Cell Genomics also addresses scTWAS. It would be useful for the authors to briefly discuss similarities and differences with that work in the discussion.

4. In Section 4.1 of the Methods, it would be beneficial to redefine all parameters, even if they were already introduced earlier in the results.

5. I think it is important to make the real data analysis code publicly available so that fellow researchers can potentially replicate the results. I also recommend making the R scripts for the method itself publicly accessible.

(Remarks on code availability)

I think the real data application codes are missing.

Version 1:

Reviewer comments:

Reviewer #1

(Remarks to the Author)

My comments have been addressed.

(Remarks on code availability)

Reviewer #2

(Remarks to the Author)

I thank the authors for their prompt and thorough responses. All of my previous comments have been satisfactorily addressed. I only note a remaining minor point of clarification regarding the definitions of z_i and s_i .

In line 115, the manuscript defines z_i as the “underlying expression level,” specified as the abundance of mRNA molecules for the gene in individual i . In line 129, s_i is defined as the sequencing depth for individual i , defined the total observed counts across all genes in the cell type of interest. The authors then assume the measurement model

$x_i | z_i \sim \text{Poisson}(s_i * z_i)$,

citing Sarkar et al. (Nature Genetics, 2021).

This formulation may cause confusion, because in Sarkar et al., the corresponding parameter λ_{ij} is defined as a relative expression level, $\lambda_{ij} = m_{ij} / m_{i+}$, rather than an absolute molecular abundance. If z_i is intended to represent an absolute molecule count, multiplying it by s_i —which is itself defined as the total observed count—would not be coherent dimensionally or conceptually.

It would therefore be helpful if the authors could clarify whether z_i is intended to represent an absolute abundance or a relative (compositional) expression level, and to adjust the notation or definitions accordingly.

No further review is required from me.

(Remarks on code availability)

Looks good to me.

Point-by-Point Response to Reviewers

We appreciate the insightful, constructive, and helpful comments from the two reviewers. Accordingly, we have made concerted efforts to carefully address each question raised as detailed below. The new or modified parts are highlighted in blue in the main text and Supplementary.

1 Reviewer 1

Introductory comment: *“The authors developed a single-cell transcriptome-wide association study (scTWAS) method based on an expression-measurement model to predict cell-type-specific (CTS) genetically regulated gene expression (GReX) and test its association with phenotype. They assume the pseudobulk RNA read counts follow a Poisson distribution with the true underlying expression as a latent variable, which is modeled as a function of genotype without assuming a known distribution. Then, a moment-based method is used for estimation. The authors conducted extensive simulations and real data applications to show that scTWAS outperforms two existing methods. The paper is well written, and the method is a timely contribution to the field. Below, I list some comments that may further improve the manuscript.”*

Reply: We would like to thank the reviewer for their encouraging comment! We have made every effort to address the comments raised in the following point-by-point responses.

1. *“The proposed scTWAS and the two existing methods seem to have two differences in GReX prediction: 1) data normalization and transformation, and 2) the variance weighted regression in Eqn 6. My impression is that the main improvement comes from the first component (expression-measurement model). scTWAS needs the variance weighting since it directly models the read counts and may be impacted by skewed values. Can the authors comment on this?”*

Reply: We agree with and appreciate this insightful comment. scTWAS directly models UMI counts through an expression-measurement model that captures heteroskedastic

technical noise and separates it from the genetically regulated component of gene expression. By addressing measurement noise explicitly, this model enables more accurate estimation of genetic effects and contributes substantially to the improved performance of scTWAS relative to the two existing approaches.

Within this expression-measurement framework, we further introduce the variance-weighted regression in Eqn 6 to achieve statistically efficient estimation. Because UMI counts exhibit heteroskedastic noise, weighting observations by the inverse of their variance naturally down-weights highly skewed or noisy values with large variance.

Taken together, the expression-measurement model provides the theoretical foundation for handling noise in single-cell data, while the variance-weighted regression offers a practical tool for efficient estimation and accurate prediction. Their *complementary* combination leads to the improved performance of scTWAS compared with existing methods that rely on heuristic normalization and unweighted regression.

To make the contribution of these two components clearer, we have added brief notes to Lines xx-xx and Line xx in the revised manuscript.

2. *“There is a competing method, scPrediXcan, first posted on bioRxiv in 2024 and published recently. The authors may discuss or compare scTWAS’s performance with the deep-learning-based scPrediXcan.”*

Reply: Thank you for pointing out this related work. This is an important and valuable development in the field, and we view it as addressing a complementary application relative to scTWAS. To clarify the respective contributions of scTWAS and scPrediXcan [1], we have added the following paragraph to the Discussion section of the revised manuscript (Lines 550-572):

Specifically, scPrediXcan leverages Enformer [2], a deep learning-based sequence-to-function model, to generate in-silico cell-type-specific gene expression for TWAS analysis. Unlike traditional TWAS methods, scPrediXcan does not use paired genotype and gene expression data across individuals to estimate genetic effects. Instead, it leverages Enformer-derived epigenomic features to generate in-silico cell-type-specific gene

expression from DNA sequence alone, and then performs TWAS using these in-silico expression profiles. While scPrediXcan offers an exciting opportunity to extend TWAS to cell types lacking population-level single-cell data, performing TWAS using in-silico expression introduces important considerations. Because the predicted expression reflects regulatory relationships imputed by a deep learning model, rather than the observed genotype-driven variation in gene expression across individuals, the resulting associations may capture model-driven regulatory patterns. Recent studies have shown that these patterns may have limited consistency with observed genotype-expression associations across individuals [3, 4]. In contrast, scTWAS directly uses paired genotype and scRNA-seq count data and a statistically principled framework to learn genotype-driven variation in gene expression across individuals. Thus, the two methods are designed for different applications. When population-scale single-cell data are not available for the cell type of interest, scPrediXcan provides sequence-based regulatory predictions, though its interpretation depends on the deep learning model’s accuracy in predicting epigenomic features across individuals. scTWAS directly leverages population-scale single-cell data, which are increasingly available for diverse tissues and diseases [5–7], to learn the observed association between genotype and gene expression in cell types.

3. *“The bulk TWAS is known to have a low out-of-sample R^2 for GReX prediction. Is the prediction accuracy improved for CTS TWAS?”*

Reply: We thank the reviewer for this insightful question. To address this question, we have conducted an additional analysis comparing the out-of-sample prediction accuracy between bulk TWAS and cell-type-specific (CTS) TWAS models using real data from the OneK1K study. Specifically, we generated a pseudobulk dataset by *aggregating all cell types* together and applied the same normalization procedure used in AN-TWAS [8], which is commonly used in bulk TWAS studies. We then compared the Stage 1 out-of-sample R^2 from the pseudobulk (representing bulk TWAS) with the corresponding cell-type-specific pseudobulk models used in CTS TWAS.

As shown in Figure R1, bulk TWAS generally exhibited higher out-of-sample R^2 than

CTS TWAS across genes, especially in less abundant cell types, which is expected given the lower sparsity and larger sequencing depth of aggregated pseudobulk data. However, we also observed a subset of genes for which CTS TWAS achieved a higher R^2 in CTS TWAS than in bulk TWAS, suggesting that the regulatory effects of these genes may be highly cell-type-specific. These findings highlight the *complementary* nature of CTS TWAS, which can capture genetic regulations that are obscured in bulk-level analyses. In general, the relative magnitude of R^2 between bulk TWAS and CTS TWAS depends on two factors: 1. the signal-to-noise (SNR) ratio of gene expression data (as determined by sample size and sparsity), and 2. the true heritability h^2 of gene expression. h^2 serves as an upper bound for R^2 : with a high SNR, h^2 will approximate R^2 , while with a low SNR (high sparsity or small sample size), R^2 can be much smaller than h^2 . In the currently available population-scale single-cell data, the SNR is still much lower than bulk data (due to higher sparsity and moderate sample size). Hence, we suspect that the low R^2 of CTS models in Figure R1 is mostly caused by the low SNR ratio of single-cell data. In addition, prior literature has suggested that the genetic regulation of gene expression may be highly CTS [9], implying higher h^2 in cell types than in bulk tissues for CTS genes. As a result, we expect that for those CTS genes, CTS R^2 will increase as more population-scale single-cell studies become available. Systematically comparing R^2 between cell types and bulk tissues while accounting for these factors is an important direction that requires a separate and in-depth investigation. We plan to explore this in future work.

4. *“APOE is the most established AD risk gene. For CTS-TWAS of AD using GWAS (Jansen et al., Nature Genetics 2019) and snRNA-seq-based eQTL summary statistics (Bryois et al., Nature Neuroscience 2022) with FUSION, APOE is significant in microglia after Bonferroni adjustment. The authors may comment on why APOE is not identified here.”*

Reply: Thank you for this insightful question! We examined our results on CTS TWAS of AD, and found that based on scTWAS’s Stage 2 p -values, APOE is significant in

Figure R1. Comparison of out-of-sample R^2 between CTS TWAS (y-axis) and bulk TWAS (x-axis).

astrocyte and 14 cell subtypes (p -value < 0.05), including proliferate, surveilling, and reacting microglia. However, its Stage 1 prediction p -values are not significant in these contexts. Following the common practice in TWAS literature [10, 11], we only report the genes with both significant Stage 1 and Stage 2 p -values as TWAS genes. As a result, scTWAS's result in the manuscript does not contain APOE as a significant TWAS genes. Similar observations hold for the results from AN-TWAS and NA-TWAS. These results suggest the presence of genetic regulation of APOE in these cellular contexts. However, in this particular single-cell dataset, APOE expression is difficult to predict, likely due to high sparsity, technical noise, and limited sample size, resulting in insufficient statistical power even when using scTWAS framework. This further underscores the importance of improving statistical modeling in Stage 1 GReX prediction, as which is critical for enhancing TWAS discovery yield.

In future studies, we will explore more strategies to further improve Stage 1 GReX prediction, such as sharing information across cell types, incorporating functional annotations, and integrating more single-cell datasets. These have the potential to uncover

APOE and more genes as underlying the genetic basis of AD in specific cell types and subtypes.

To address this comment, we have discussed this result on Lines 482-488 of the revised manuscript.

2 Reviewer 2

Introductory comment: *“Lin and Su proposed a novel method, scTWAS, to extend the traditional TWAS framework to scRNA-seq settings. I found the method interesting and applaud the authors for conducting extensive real data applications. My major concern is whether the simulations reflect real-world scenarios. I’ve listed my major and minor comments below: ”*

Reply: We would like to thank the reviewer for their encouraging comment! We have made every effort to address the comments raised in the following point-by-point responses.

Major Comments:

1. *“In the scTWAS scenario, I think the most interesting question is: given a trait, are the scTWAS-identified genes more cell-type-specific or more shared across cell types? Based on the real data results, can the authors comment on this question or try to quantify it? ”*

Reply: Thank you for raising this important question. In the original manuscript, we examined the degree to which scTWAS-identified genes of the three immune-related diseases are cell-type-specific versus shared across immune cell types in section 2.5 (now Lines 381-411), and we also visualized the results using UpSet plots (now Supplementary Figure S16). Specifically, for rheumatoid arthritis, 21 of 64 significant genes (33%) were specific to a single cell type; for SLE, 20 of 41 genes (49%) were cell-type-specific; and for asthma, 57 of 111 genes (51%) were identified in only one cell type. We also discuss several factors that can give rise to cell-type-specific TWAS associations, including expression restricted to a single cell type (e.g., *ZFP57* in CD4_{NC}), cell-type-specific

genetic regulation that leads to predictive GReX models only in a specific cell type (e.g., *FLOT1* in Mono_{NC}), and trait associations that are specific to a cell type despite broader predictability (e.g., *PSMB9* for RA in CD8_{ET}). To make this clearer to readers, we now explicitly summarize the quantitative patterns in the main text (Lines 381-411), accompanying the visualization in Figures 4d-g.

For Alzheimer’s disease (AD), in our original manuscript, we visualize the specificity of scTWAS-identified genes across microglia subtypes in Figure 5b, where we can see that most of the genes were shared across at least two microglia subtypes, and only six genes were specific to a single subtype. In addition, we have now added a *new* investigation for the specificity of scTWAS-identified genes across *major cell types* using ROSMAP brain scRNA-seq dataset. As shown in Figure R2 (Supplementary Figure S17, line 455), most AD-associated genes identified by scTWAS were specific to a major brain cell type.

Figure R2. **UpSet plot of scTWAS-identified AD genes across all major cell types.** Results are aggregated to the major cell type level.

We note that in general, the apparent cell-type-specificity of scTWAS-identified genes can vary for several reasons. It depends on the similarity among cell types (for example, scTWAS-identified genes appear less distinct across microglia subtypes (Figure 5b) than

across major brain cell types (Figure R2), the sample size available for each cell type (a cell type with true genetic regulation may appear insignificant due to limited sample size, leading to apparent specificity), and other biological factors.

Moreover, while marginal TWAS significance across cell types provides an initial view of similarity versus specificity, it can be difficult to interpret directly because cell types often share regulatory architectures, and correlated genetic effects or LD can produce apparent overlap even when only one cell type is truly relevant. For these reasons, we view the systematic investigation of cell-type-specificity of gene-trait associations as a nontrivial and important direction that requires a separate and dedicated study. A natural approach is to incorporate fine-mapping to evaluate whether a gene-trait association is specific to a cell type or shared across multiple cell types, drawing inspiration from recent work on fine-mapping gene-tissue pairs [12, 13]. Integrating such cell-type-aware fine-mapping approaches with scTWAS findings could enhance the resolution and interpretability of scTWAS signals. This will be a focus of our future methodological development. We have now added this to the Discussion section (Lines 576-585).

2. *“In the introduction, the authors mentioned both cell types and cell subtypes. I think it’s necessary to explicitly define what is meant by these terms. As a result, I found the statement “the heterogeneity across cell subtypes will be obscured” a bit confusing. That being said, can the authors also clarify how their method disentangles heterogeneity across cell subtypes? Please correct me if I’ve misunderstood anything here.”*

Reply: We thank the reviewer for this helpful clarification request. We re-examined the wording in the Introduction and agree that the terminology could be made clearer. We have revised the text accordingly (Lines 43-44, 55).

In our manuscript, we use cell type to denote a broad cellular category (e.g., microglia) and cell subtype to denote a more refined grouping within a cell type (e.g., homeostatic or inflammatory microglia), defined by distinct gene expression signatures.

In this sense, both cell types and cell subtypes can be viewed as cell groups, with

cell types representing a coarser grouping and cell subtypes providing a more refined resolution. scTWAS is agnostic to this level of granularity: depending on the biological question, one may apply scTWAS to cells belonging to a major cell type or to a specific subtype to identify TWAS genes within that group. This allows heterogeneity of TWAS associations to be examined at either the cell-type or cell-subtype level.

Finally, we would like to clarify our intended meaning of the statement “the heterogeneity across cell subtypes will be obscured”: this sentence refers to a limitation of using cell sorting for obtaining cell-type-specific gene expression profiles, which typically rely on well established markers for major cell types [14, e.g.] and therefore can primarily isolate broad cell type categories. In comparison, markers for refined cell subtypes are less well studied, and thus it is technically more challenging to use cell sorting to study gene expression and genetic regulation in cell subtypes. We have modified this sentence in the main text (Line 55) to clarify this point.

3. *“In the Methods section, it’s not clear to me how the genotype data were simulated. Did the simulation reflect the real LD structure across SNPs within a gene window? The manuscript mentions that “10 causal cis-SNPs were randomly selected.” Ten seems like a large number, since most genes only have 1-3 cis-eQTLs. While regularized prediction models often select more SNPs with non-zero weights (e.g., around 10), that reflects predictive modeling, not the true number of causal eQTLs. Strictly speaking, TWAS tests the association between predicted gene expression and the trait—not the actual expression levels—based on the current two-step TWAS framework.”*

Reply: Thank you for the question! We apologize for the lack of clarity in the original Methods description. In our simulation, the genotype data were not generated synthetically; rather, we directly used the real genotype data from the OneK1K study. Specifically, for each of the 20 randomly selected genes, we extracted the observed genotypes of all *cis*-SNPs (500 kB upstream of the gene transcription start site and 500 kB downstream of the transcription end site) from 982 individuals in the OneK1K

cohort. Therefore, the simulated data fully preserved the real LD structure within each gene window.

We agree with the reviewer that 10 causal *cis*-SNPs may be a large number. To reflect more realistic genetic architectures, we have now added an additional simulation scenario in which only 3 causal *cis*-SNPs were selected for a gene. As shown in Figure R3a-c, we observed the same conclusion that scTwas consistently achieved better Stage 1 prediction accuracy, higher Stage 2 power than the other two existing methods, and maintained appropriate type-I error control. For completeness, and in response to Reviewer 2’s comment 5, the results in Figure R3b-c also consider different GWAS sample sizes.

These new results are provided in Supplementary Figure S6 and discussed in the revised manuscript (Lines 214-225).

We also agree with the reviewer that “TWAS tests the association between predicted gene expression and the trait – not the actual expression levels – based on the current two-step TWAS framework“. To make this more rigorous, we clarified this in the revised manuscript (Lines 91, 111, 141, 643).

Figure R3. **Simulation results with three causal *cis*-SNPs under a two-sample design.** **a.** Proportion of predictive GREX models with nominal p -value < 0.05 . **b.** Empirical power when $h_g^2 = 0.2/180$. **c.** Empirical type-I error when $h_g^2 = 0$. Stage 1 models were trained with $N_{eQTL} = 982$.

4. “ In Section 4.4, several parameters and symbols are not defined, so I’m not 100%

sure I understood it correctly. Does y represent the complex trait? Does h_g^2 represent the trait heritability explained by a single gene? If so, setting $h_g^2 = 0.05$ seems very high. For example, if trait heritability is 0.3 and there are 1000 causal genes, per-gene heritability would be $0.3 / 1000 = 3e-4$.”

Reply: Thank you for this careful and helpful comment. Yes, y represents the complex trait and h_g^2 represents the trait heritability explained by a single gene in our original simulation design. We have now clarified this in Lines 744-746 and 753-756 of the revised main text.

We fully agree that the value $h_g^2 = 0.05$ used in the initial setup is quite high. This choice was made to ensure nontrivial power for Stage 2 association testing given the small GWAS sample size ($N_{\text{GWAS}} = 982$) in our original one-sample TWAS simulation design. To address this concern, we have now added a new simulation setup where we varied the per-gene heritability over realistic values and considered a two-sample simulation design with larger GWAS sample sizes (Reviewer 2, comment 5). Following the design used in [11], we considered a heritable phenotype in which gene expression explains 0.10/180 or 0.20/180 of the phenotypic variance, corresponding to typical and high-effect loci reported in large-scale GWAS, respectively. We additionally included a setting of $h_g^2 = 0.05/180$ for low-effect loci.

Figure R4 shows the result across different h_g^2 and for completeness, these results also incorporate different GWAS sample sizes (Reviewer 2, comment 5). As expected, power increases with larger per-gene heritability, and the relative improvement of scTWAS compared with existing approaches remains consistent across settings. The new results are provided in Supplementary Figure S5 and discussed in the revised manuscript (Lines 214-225).

Figure R4. Empirical power to detect gene-trait associations across different proportions of trait heritability explained and GWAS sample sizes (from top to bottom: 30,000, 50,000, 100,000) for three cell types (columns). The simulation uses genotype data from 982 individuals in the OneK1K cohort and considers three representative immune cell types.

5. “In the Methods section, it’s not clear how the GWAS data were simulated. Did the authors account for LD? How was GWAS sample size incorporated? Also, in practice, eQTL and GWAS samples are from different cohorts. Was this reflected in the simulation design?”

Reply: Thank you for the questions! In the original simulation setup, both gene expression and complex trait phenotypes y were generated using the genotype data of 982 individuals from the OneK1K cohort, which captures the LD structure in real data. We then fit the Stage 1 to get $\hat{\beta}$, and evaluated Stage 2 by testing the association between y and $G\hat{\beta}$ using individual-level data. This represents a one-sample TWAS

design where both eQTL and GWAS samples are from the same cohort.

We agree that our original simulation does not fully capture the complexity of TWAS analyses in practice. To address the Reviewer 2's questions, we have added a new simulation study that reflects a two-sample TWAS design using only GWAS summary statistics and considers varying eQTL and GWAS sample sizes. In this new setup, we continue to use the OneK1K data to generate gene expression (Stage 1), preserving the real LD structure. We consider two eQTL sample sizes, either using all 982 OneK1K individuals or using a reduced random sample of 500 individuals. For Stage 2, instead of simulating individual-level trait data from the same cohort, we directly generate GWAS summary statistics (the **marginal** SNP-trait association estimates and corresponding standard error), representing a two-sample design for TWAS analyses. Specifically, for a given pair of gene and cell type, we computed the per-SNP genotype standard deviations $D_{jj} = \sqrt{2p_j(1-p_j)}$, where p_j is the minor allele frequency, and the LD matrix \mathbf{V} from the OneK1K genotypes. Given the true (joint) SNP effect on the GWAS trait \mathbf{b}_{GWAS} , the observed marginal GWAS summary statistics were generated as follows [15]:

$$\hat{\boldsymbol{\beta}}_{\text{GWAS}} \sim \mathcal{N}\left(\mathbf{D}^{-1}\mathbf{V}\mathbf{D}\mathbf{b}_{\text{GWAS}}, (\sigma^2/N_{\text{GWAS}})\mathbf{D}^{-1}\mathbf{V}\mathbf{D}^{-1}\right), \quad \text{SE}_j = \frac{\sigma}{\sqrt{N_{\text{GWAS}}2p_j(1-p_j)}}, \quad (\text{R1})$$

where $\mathbf{D} = \text{diag}(D_{jj})$, N_{GWAS} is the sample size of GWAS, and σ is the residual standard deviation of the GWAS trait. To generate the joint SNP effect on the GWAS trait \mathbf{b}_{GWAS} , we set $\mathbf{b}_{\text{GWAS}} = \gamma\boldsymbol{\beta}$, where $\boldsymbol{\beta}$ is the SNP genetic effect on the gene expression. We then chose $\gamma = \sqrt{h_g^2/\text{Var}(\mathbf{G}\boldsymbol{\beta})}$, where $\text{Var}(\mathbf{G}\boldsymbol{\beta}) = \boldsymbol{\beta}'\mathbf{D}\mathbf{V}\mathbf{D}\boldsymbol{\beta}$, so that the trait heritability explained by the genetically regulated gene expression is h_g^2 , and $\sigma^2 = 1 - h_g^2$. This framework allows us to vary the GWAS sample size while ensuring that LD among SNPs is correctly accounted for.

Figure R5 (now Figure 1c in the main text) shows the empirical power under the new two-sample TWAS simulation framework with varying eQTL sample sizes (500 vs. 982) and GWAS sample sizes (30,000, 50,000, 100,000). Across all three representative cell types, power increased with larger GWAS sample sizes and with larger eQTL sample sizes,

as expected. Importantly, scTWAS consistently achieved higher power than the two existing approaches. These results confirm that scTWAS maintains its power advantage under realistic simulation settings that emulate two-sample TWAS analyses based on GWAS summary statistics.

The new simulation design is now described in Section 4.4.2. A summary of the results is provided in the main text (Lines 198-207, Figure 1c), and the full set of results is presented in Supplementary Figures S3 and S5.

Figure R5. Empirical power from two-sample TWAS simulation across different expression reference panel sample sizes (top: 500, bottom: 982) and GWAS sample sizes. The simulation uses OneK1K genotype data and considers three representative immune cell types. The gene’s *cis*-SNPs are set to explain 0.1/180 of the trait variance.

6. *“I think some additional simulations are needed, such as: Varying per-gene heritability with more realistic values, Simulating both in-sample and out-of-sample LD, Considering pleiotropic effects of cis-eQTLs, Varying eQTL and GWAS sample sizes, Model misspecification cases, such as when the selected cell type doesn’t match the cell type relevant to the trait.”*

Reply: Thank you for these excellent suggestions! We have incorporated these recommendations into our revised simulation study. Under the updated two-sample TWAS simulation framework described above, where Stage 1 uses real OneK1K genotypes to generate single-cell gene expression and Stage 2 directly simulates GWAS summary statistics, we have now performed the following additional simulation studies:

(a) **Varying eQTL sample sizes, GWAS sample sizes, and per-gene heritability.** We first varied eQTL sample sizes from 500 to 982 individuals and evaluated the performance of Stage 1 prediction models. Figure R6 shows that the improvement of Stage 1 prediction by scTwas over existing methods remains consistent across eQTL sample sizes and cell types. Figure R7 further presents the Stage 2 power of three methods across eQTL sample sizes, GWAS sample sizes, and per-gene trait heritability. Across all scenarios, scTwas consistently outperformed both NA-Twas and AN-Twas in yielding higher power to detect true gene-trait associations. These new results are now presented in Supplementary Figure S5.

Figure R6. Proportion of predictive GReX models with nominal p -value < 0.05 in two-sample simulations across $N_{eQTL} \in \{500, 982\}$.

Figure R7. Empirical power to detect gene-trait associations in two-sample simulations across different per-gene trait heritability (0.05/180, 0.1/180, 0.2/180), expression reference panel sample sizes (500 and 982), GWAS sample sizes (3×10^4 , 5×10^4 , 10^5), and three cell types.

- (b) **Simulating LD mismatch (out-of-sample LD).** To evaluate the robustness of scTWAS to LD mismatch, we simulated GWAS summary statistics using the true LD matrix estimated from OneK1K and then performed Stage 2 TWAS using an out-of-sample LD reference. The out-of-sample LD matrix \mathbf{V}_{ref} was generated by first drawing a sample covariance matrix \mathbf{S} from a Wishart distribution centered at a slightly regularized version of the true LD \mathbf{V}_λ (Eq. (R2)), mimicking the finite-sample variation encountered when using an external LD panel of the same ancestry

$$\mathbf{S} = \frac{1}{\nu} \mathbf{W}, \quad \mathbf{W} \sim \text{Wishart}(\nu, \mathbf{V}_\lambda), \quad \mathbf{V}_\lambda = (1 - 10^{-3})\mathbf{V} + 10^{-3}\mathbf{I} \quad (\text{R2})$$

with $\nu = 2000$ representing the effective LD reference size. We then set $\mathbf{V}_{\text{ref}} = \text{cov2cor}(\mathbf{S})$. As shown in Figure R8a, TWAS power using this out-of-sample LD

reference was nearly identical to those obtained using the in-sample LD matrix (Figure R5 and Figure 1c in the main text), demonstrating that scTWAS is robust to realistic levels of LD mismatch. In addition, Figure R8b shows that Type-I error rates under out-of-sample LD were also well controlled.

Figure R8. Simulation results with an out-of-sample LD in Stage 2 and $N_{eQTL} = 982$ under a two-sample design. **a.** Empirical power when $h_g^2 = 0.2/180$. **b.** Empirical type-I error when $h_g^2 = 0$. Stage 1 models were trained with $N_{eQTL} = 982$.

(c) **Considering pleiotropic effects of *cis*-SNPs.** To evaluate the impact of horizontal pleiotropy, we extended the simulation to allow a subset of *cis*-SNPs to exert direct effects on the trait independent of gene expression. Specifically, we set the true SNP-trait effects in Eq. (R1) to $\mathbf{b}_{GWAS} = \gamma\boldsymbol{\beta} + \boldsymbol{\alpha}$, where $\boldsymbol{\beta}$ are the SNP genetic effects on gene expression, and $\boldsymbol{\alpha}$ captures pleiotropic effects from 2 of the causal SNPs, scaled by to explain $k \times (0.2/180)$ of the trait variance ($k \in \{0.5, 1\}$). We evaluated both the type-I error ($h_g^2 = 0$) and power ($h_g^2 = 0.2/180$) of three methods. As shown in Figure R9, all three TWAS approaches exhibited type-I error inflation under horizontal pleiotropy, with stronger inflation at larger k . When type-I error is estimated based on Stage 2 p -values (Figure R9a, left column), the inflation is similar across methods, suggesting that all methods are comparably susceptible to horizontal pleiotropy. Although scTWAS appears more

inflated when significant discoveries are further conditioned on having a predictive Stage 1 model (Figure R9b, left column), this reflects that scTWAS more often produces a predictive Stage 1 model, rather than more inflated Stage 2 p -values due to horizontal pleiotropy. We also applied LDA-Egger [16], which reduced type-I error but substantially lowered power (Figure R10). Overall, these results confirm that all three TWAS methods, including scTWAS, are susceptible to horizontal pleiotropy, consistent with known limitations of standard TWAS. We recognize this as an important challenge and plan to address it in future work.

We have now incorporated these results in Supplementary Section S1.3.2, and briefly discussed them in the main text (Lines 222-225).

Figure R9. Simulation results for MonoC in the presence of horizontal pleiotropy. **a**-**b**. Empirical type-I error rate when $h_g^2 = 0$ and power when $h_g^2 = 0.2/180$ across k (the strength of pleiotropic effect) and GWAS sample sizes. In **a**., a significant Stage 2 association was counted as long as Stage 2 test was significant; whereas in **b**., a significant Stage 2 association was counted when the gene had both a predictive Stage 1 model (with prediction p -value < 0.05) and a significant Stage 2 test.

Figure R10. LDA-Egger results for MonoC in the presence of horizontal pleiotropy. **a-b.** Empirical type-I error rate when $h_g^2 = 0$ and power when $h_g^2 = 0.2/180$ across k (the strength of pleiotropic effect) and GWAS sample sizes. In **a.**, a significant Stage 2 association was counted as long as Stage 2 test was significant; whereas in **b.**, a significant Stage 2 association was counted when the gene had both a predictive Stage 1 model (with prediction p -value < 0.05) and a significant Stage 2 test.

(d) **Considering model misspecification.** When the tested cell type A is not the true trait-relevant cell type B, and the two cell types do not share similar genetic effects on the tested gene, the expected true TWAS effect is zero; this scenario is covered by our type-I error evaluation (Figure R11 and Supplementary Table S1). More complicated mismatch settings, such as identifying causal cell types in the presence of genetic co-regulation across multiple cell types, would require a multivariable TWAS framework and are beyond the scope of the current manuscript. We plan to explore this, together with the cell-type-specificity of scTWAS findings (Reviewer 2, comment 1) in a separate project. To acknowledge this important direction, we have added a remark to the Discussion section on Lines 576-585.

Figure R11. Empirical type-I error rates of gene-trait association tests in two-sample simulations across $N_{eQTL} \in \{500, 982\}$ and $N_{GWAS} \in \{3 \times 10^4, 5 \times 10^4, 10^5\}$.

7. *“In the real data analysis, especially in visualizations and comparisons, I think the authors should always include conventional bulk RNA-seq TWAS as a baseline. This would give readers a better sense of how much is gained by incorporating scRNA-seq data.”*

Reply: Thank you for the great suggestion! In our original real-data analysis of 29 hematological traits, we compared our cell-type-level TWAS findings with conventional bulk TWAS results from Rowland et al. [17], which used bulk whole blood RNA-seq data from 922 European individuals to train GReX models and the same 29 GWAS data from UK Biobank. As shown in Figure 4c, the cell-type-level TWAS uniquely identified novel loci that were missed by the bulk tissue-level TWAS, with scTWAS detecting the largest number of novel associations overall. We also discussed two specific examples, *HHEX* and *DGCR6*, in Lines 309-333.

Following the reviewer’s recommendation, we have now added an additional analysis to provide a more direct baseline comparison using the same dataset as our analysis. Specifically, we generated a new pseudobulk dataset by aggregating all cells across cell types and applied the same normalization procedure used in AN-TWAS. Using this new

pseudobulk expression as the bulk RNA-seq data, we retrained bulk GReX prediction models and compared their Stage 1 and Stage 2 performance with the CTS TWAS models. We use this analysis to represent the performance of conventional bulk TWAS below.

We observed that, in the analysis of OneK1K data, CTS TWAS achieves better prediction performance for a subset of genes (Figure R12) and identifies novel gene-trait associations (Figure R13) in immune cell types that are not identified in bulk analyses, highlighting the unique insights gained by performing cell-type-level TWAS with scRNA-seq data. Similar patterns are observed in the ROSMAP analysis of brain tissues (Figure R14). At the same time, we acknowledge that bulk TWAS may achieve better Stage 1 prediction performance (Figure R1), which is expected given the lower sparsity and higher sequencing depth of bulk data used in this analysis.

In summary, we have added new analysis and comparison to quantify the gain by scTWAS with scRNA-seq data compared to conventional bulk TWAS, using either external bulk RNA-seq TWAS, or a pseudo-bulk approach that emulates bulk data using scRNA-seq data. Across both scenarios and across blood and brain tissues, we demonstrate the unique findings by scTWAS compared to bulk TWAS, highlighting the benefits of incorporating scRNA-seq data. We also acknowledge that, due to the limited sample size and the high sparsity of currently available scRNA-seq data, we do not expect the cell-type-level TWAS to yield more discoveries across all genes and cell types. As a future direction, we will also explore integrating bulk data to borrow information and help improve the cell-type-level TWAS with scRNA-seq data.

We have incorporated a brief summary of this comparison into the revised Results section (Lines 275-284, 333-337, 378-380, 467-469) and Supplementary Figures S13-15.

Figure R12. Number of genes with significantly predictive Stage 1 GR_{EX} models in cell-type-specific TWAS that were missed by bulk-like TWAS across 13 immune cell types.

Figure R13. Number of novel TWAS-identified genes identified by cell-type-specific TWAS that were missed by bulk-like TWAS, across 29 hematological traits (left) and 3 immune-related diseases (right).

Figure R14. Comparison between cell-subtype-specific TWAS and bulk-like TWAS in ROSMAP study. **a.** Number of genes with significantly predictive Stage 1 GReX models in brain cell subtypes that were missed by the bulk-like TWAS. **b.** Number of TWAS genes in brain cell subtypes that were missed by the bulk-like TWAS. Subtype counts are aggregated within each major cell type.

8. *“In the discussion of limitations and future directions, the authors may want to mention the potential extension to fine-mapping and multi-ancestry settings, along with relevant references. Also, Wainberg et al., Nature Genetics, 2019 is an important citation when discussing limitations of TWAS methods.”*

Reply: Thank you for the suggestion. We have revised the Discussion section to highlight these important future directions. Specifically, we now mention the potential extension of scTWAS to incorporate fine-mapping strategies that better distinguish causal genes and causal cell types from LD-linked associations. We have also expanded our discussion to include multi-ancestry applications. As recommended, we now cite [18] in our discussion of TWAS limitations. These revisions appear in Lines 576-585 and 596-601 of the Discussion.

Minor Comments:

1. *“In the abstract, the use of the word “overlooking” seems subjective. I don’t think it’s fair to say that TWAS overlooked cell types; rather, the necessary data simply weren’t available when TWAS was first developed.”*

Reply: Thank you for this comment. We have now revised the abstract to use more

neutral wording, which now states, "...but most have been performed using bulk gene expression data, which aggregate signals across heterogeneous cell types."

2. *"It would be helpful if the authors could add line numbers to the manuscript so reviewers can reference specific sections more easily."*

Reply: Thank you for the suggestion! We have now added line numbers to the manuscript.

3. *"Recently, a similar paper by Zhou Y. et al., Cell Genomics also addresses scTWAS. It would be useful for the authors to briefly discuss similarities and differences with that work in the discussion."*

Reply: Thank you for pointing out this related work. We have now added a paragraph in the Discussion comparing scTWAS with scPrediXcan (Lines 550-572). Specifically, scPrediXcan leverages Enformer [2], a deep learning-based sequence-to-function model, to generate in-silico cell-type-specific gene expression for TWAS analysis. Unlike traditional TWAS methods, scPrediXcan does not use paired genotype and gene expression data across individuals to estimate genetic effects. Instead, it leverages Enformer-derived epigenomic features to generate in-silico cell-type-specific gene expression from DNA sequence alone, and then performs TWAS using these in-silico expression profiles. While scPrediXcan offers an exciting opportunity to extend TWAS to cell types lacking population-level single-cell data, performing TWAS using in-silico expression introduces important considerations. Because the predicted expression reflects regulatory relationships imputed by a deep learning model, rather than the observed genotype-driven variation in gene expression across individuals, the resulting associations may capture model-driven regulatory patterns. Recent studies have shown that these patterns may have limited consistency with observed genotype-expression associations across individuals [3, 4]. In contrast, scTWAS directly uses paired genotype and scRNA-seq count data and a statistically principled framework to learn genotype-driven variation in gene expression across individuals. Thus, the two methods are designed for different applications. When population-scale single-cell data are not available for the cell type

of interest, scPrediXcan provides sequence-based regulatory predictions, though its interpretation depends on the deep learning model's accuracy in predicting epigenomic features across individuals. scTWAS directly leverages population-scale single-cell data, which are increasingly available for diverse tissues and diseases [5–7], to learn the observed association between genotype and gene expression in cell types.

4. *“In Section 4.1 of the Methods, it would be beneficial to redefine all parameters, even if they were already introduced earlier in the results.”*

Reply: We thank the reviewer for this suggestion. We have revised Section 4.1 to redefine all parameters and notations directly in the Methods, ensuring that the section is self-contained and does not rely on earlier definitions. These changes appear in Lines 612-618.

5. *“I think it is important to make the real data analysis code publicly available so that fellow researchers can potentially replicate the results. I also recommend making the R scripts for the method itself publicly accessible.”*

Reply: Thank you for the suggestion. We agree that it is important to make the code publicly available to enable replications and applications by fellow researchers. We have now added a section called “Code availability” and published our code for both real data analysis and method implementation on GitHub. The R package for scTWAS is available at <https://github.com/ZhaotongL/scTWAS>, and the real data analysis code is available at https://github.com/ZhaotongL/scTWAS_paper.

References

- [1] Zhou, Y. *et al.* scpredixcan integrates deep learning methods and single-cell data into a cell-type-specific transcriptome-wide association study framework. *Cell Genomics* **5** (2025).

- [2] Avsec, Ž. *et al.* Effective gene expression prediction from sequence by integrating long-range interactions. *Nature methods* **18**, 1196–1203 (2021).
- [3] Huang, C. *et al.* Personal transcriptome variation is poorly explained by current genomic deep learning models. *Nature Genetics* **55**, 2056–2059 (2023).
- [4] Sasse, A. *et al.* Benchmarking of deep neural networks for predicting personal gene expression from dna sequence highlights shortcomings. *Nature Genetics* **55**, 2060–2064 (2023).
- [5] Natri, H. M. *et al.* Cell-type-specific and disease-associated expression quantitative trait loci in the human lung. *Nature Genetics* **56**, 595–604 (2024).
- [6] Cuomo, A. S. *et al.* Impact of rare and common genetic variation on cell type-specific gene expression. *medRxiv* 2025–03 (2025).
- [7] Emani, P. S. *et al.* Single-cell genomics and regulatory networks for 388 human brains. *Science* **384**, eadi5199 (2024).
- [8] Zeng, L. *et al.* A single-nucleus transcriptome-wide association study implicates novel genes in depression pathogenesis. *Biological Psychiatry* **96**, 34–43 (2024).
- [9] Umans, B. D., Battle, A. & Gilad, Y. Where are the disease-associated eqtls? *Trends in Genetics* **37**, 109–124 (2021).
- [10] Gamazon, E. R. *et al.* A gene-based association method for mapping traits using reference transcriptome data. *Nature genetics* **47**, 1091–1098 (2015).
- [11] Gusev, A. *et al.* Integrative approaches for large-scale transcriptome-wide association studies. *Nature genetics* **48**, 245–252 (2016).
- [12] Strober, B. J., Zhang, M. J., Amariuta, T., Rossen, J. & Price, A. L. Fine-mapping causal tissues and genes at disease-associated loci. *Nature Genetics* **57**, 42–52 (2025).
- [13] Yang, Y., Lorincz-Comi, N. & Zhu, X. Uncovering causal gene-tissue pairs and variants through a multivariate twas controlling for infinitesimal effects. *Nature Communications* **16**, 6098 (2025).

- [14] Schmiedel, B. J. *et al.* Impact of genetic polymorphisms on human immune cell gene expression. *Cell* **175**, 1701–1715 (2018).
- [15] Zhu, X. & Stephens, M. Bayesian large-scale multiple regression with summary statistics from genome-wide association studies. *The annals of applied statistics* **11**, 1561 (2017).
- [16] Barfield, R. *et al.* Transcriptome-wide association studies accounting for colocalization using egger regression. *Genetic epidemiology* **42**, 418–433 (2018).
- [17] Rowland, B. *et al.* Transcriptome-wide association study in uk biobank europeans identifies associations with blood cell traits. *Human Molecular Genetics* **31**, 2333–2347 (2022).
- [18] Wainberg, M. *et al.* Opportunities and challenges for transcriptome-wide association studies. *Nature genetics* **51**, 592–599 (2019).

Point-by-Point Response to Reviewer 2

We sincerely thank Reviewer 2 for the additional comments, which helped clarify the definition of our model and improve the presentation of our method. Below, we reproduce the comment and provide our response.

1. *“I thank the authors for their prompt and thorough responses. All of my previous comments have been satisfactorily addressed. I only note a remaining minor point of clarification regarding the definitions of z_i and s_i . In line 115, the manuscript defines z_i as the “underlying expression level,” specified as the abundance of mRNA molecules for the gene in individual i . In line 129, s_i is defined as the sequencing depth for individual i , defined the total observed counts across all genes in the cell type of interest. The authors then assume the measurement model $x_i|z_i \text{ Poisson}(s_i * z_i)$, citing Sarkar et al. (Nature Genetics, 2021). This formulation may cause confusion, because in Sarkar et al., the corresponding parameter λ_{ij} is defined as a relative expression level, $\lambda_{ij} = m_{ij}/m_{i+}$, rather than an absolute molecular abundance. If z_i is intended to represent an absolute molecule count, multiplying it by s_i —which is itself defined as the total observed count—would not be coherent dimensionally or conceptually. It would therefore be helpful if the authors could clarify whether z_i is intended to represent an absolute abundance or a relative (compositional) expression level, and to adjust the notation or definitions accordingly. No further review is required from me.”*

Reply: We thank Reviewer 2 for this insightful comment. We agree that it is more rigorous to define z_i as the relative (compositional) expression level. To address this point, we have revised Lines 116-118 to define z_i accordingly, as follows:

“For a given gene-cell type pair, let z_i denote the underlying expression level, defined to be the relative abundance of this gene (i.e., the number of mRNA molecules from this gene relative to the total number of molecules in a cell) from this cell type in individual i .”